# Methamphetamine-induced adaptation of learning rate dynamics depend on baseline performance

Hans Kirschner[1]*, Hanna M Molla[2], Matthew R Nassar[3,4], Harriet de Wit[2†], Markus Ullsperger[1,5,6†]

[1]Institute of Psychology, Otto-von-Guericke University, Magdeburg, Germany; [2]Department of Psychiatry and Behavioral Neuroscience, University of Chicago, Chicago, United States; [3]Robert J. and Nancy D. Carney Institute for Brain Science, Brown University, Providence, United States; [4]Department of Neuroscience, Brown University, Providence, United States; [5]Center for Behavioral Brain Sciences, Magdeburg, Germany; [6]German Center for Mental Health (DZPG), Center for Intervention and Research on Adaptive and Maladaptive Brain Circuits Underlying Mental Health (C-I-R-C), Halle-Jena-Magdeburg, Germany

*For correspondence:
hans.kirschner@ovgu.de

†These authors contributed equally to this work

## eLife Assessment

This manuscript reports effects of a single dose of methamphetamine vs placebo on a probabilistic reversal learning task with different levels of noise, in a large group of young healthy volunteers. The paper is well written and the methods are rigorous. The findings are **important** and have theoretical or practical implications beyond a single a subfield. The strength of the evidence is **convincing**, with the methods, data, and analyses broadly supporting the claims in the paper, which are sufficiently qualified given the lack of a significant effect of the binary baseline performance variable, and the nonlinear effect of individual differences in baseline performance.

**Abstract** The ability to calibrate learning according to new information is a fundamental component of an organism's ability to adapt to changing conditions. Yet, the exact neural mechanisms guiding dynamic learning rate adjustments remain unclear. Catecholamines appear to play a critical role in adjusting the degree to which we use new information over time, but individuals vary widely in the manner in which they adjust to changes. Here, we studied the effects of a low dose of methamphetamine (MA), and individual differences in these effects, on probabilistic reversal learning dynamics in a within-subject, double-blind, randomized design. Participants first completed a reversal learning task during a drug-free baseline session to provide a measure of baseline performance. Then they completed the task during two sessions, one with MA (20 mg oral) and one with placebo (PL). First, we showed that, relative to PL, MA modulates the ability to dynamically adjust learning from prediction errors. Second, this effect was more pronounced in participants who performed moderately low at baseline. These results present novel evidence for the involvement of catecholaminergic transmission on learning flexibility and highlights that baseline performance modulates the effect of the drug.

## Introduction

Goal-directed behavior requires organisms to continually update predictions about the world to select actions in the light of new information. In environments that include discontinuities (changepoints) and noise (probabilistic errors), optimal learning requires increased weighting of surprising information during periods of change and ignoring surprising events during periods of stability. A burgeoning literature suggests that humans are able to calibrate learning rates according to the statistical content of new information (*Behrens et al., 2007*; *Cook et al., 2019*; *Diederen et al., 2016*; *Nassar et al., 2019*; *Nassar et al., 2010*; *Razmi and Nassar, 2022*), albeit to varying degrees (*Kirschner et al., 2022*; *Kirschner et al., 2024*; *Nassar et al., 2016*; *Nassar et al., 2012*; *Nassar et al., 2021*).

Although the exact neural mechanisms guiding dynamic learning adjustments are unclear, several neuro-computational models have been put forward to characterize adaptive learning. While these models differ in their precise computational mechanisms, they share the hypothesis that catecholamines play a critical role in adjusting the degree to which we use new information over time. For example, a class of models assumes that striatal dopaminergic prediction errors act as a teaching signal in cortico–striatal circuits to learn task structure and rules (*Badre and Frank, 2012*; *Collins and Frank, 2013*; *Collins and Frank, 2016*; *Lieder et al., 2018*; *Pasupathy and Miller, 2005*; *Schultz et al., 1997*). Another line of research highlights the role of dopamine in tracking the reward history with multiple learning rates (*Doya, 2002*; *Kolling et al., 2016*; *Meder et al., 2017*; *Schweighofer and Doya, 2003*). This integration of reward history over multiple time scales enables people to estimate trends in the environment through past and recent experiences and adjust actions accordingly (*Wilson et al., 2013*). Within the broader literature of cognitive control, it has been suggested that dopamine in the prefrontal cortex and basal ganglia is involved in modulating computational tradeoffs such as cognitive stability–flexibility balance (*Cools, 2008*; *Dreisbach et al., 2005*; *Floresco, 2013*; *Goschke, 2013*; *Goschke and Bolte, 2014*; *Goschke and Bolte, 2018*). In particular, it has been proposed that dopamine plays a crucial role in the regulation of meta-control parameters that facilitate dynamic switching between complementary control modes (i.e., shielding goals from distracting information vs. switching goals in response to significant changes in the environment) (*Goschke, 2013*; *Goschke and Bolte, 2014*; *Goschke and Bolte, 2018*). Finally, other theories highlight the importance of the locus coeruleus/norepinephrine system in facilitating adaptive learning and structure learning (*Razmi and Nassar, 2022*; *Silvetti et al., 2018*; *Yu et al., 2021*). Consistent with these neuro-computational models catecholaminergic drugs are known to affect cognitive performance including probabilistic reversal learning (*Cook et al., 2019*; *Dodds et al., 2008*; *Repantis et al., 2010*; *Rostami Kandroodi et al., 2021*; *van den Bosch et al., 2022*; *Westbrook et al., 2020*). Indeed, psychostimulants, such as methamphetamine (MA), that increase extracellular catecholamine availability, can enhance cognition (*Arria et al., 2017*; *Husain and Mehta, 2011*; *Smith and Farah, 2011*) and are used to remediate cognitive deficits in attention deficit hyperactivity disorder (*Arnsten and Pliszka, 2011*; *Prince, 2008*). However, the cognitive enhancements vary across tasks and across individuals (*Bowman et al., 2023*; *Cook et al., 2019*; *Cools and D'Esposito, 2011*; *Garrett et al., 2015*; *Rostami Kandroodi et al., 2021*; *van den Bosch et al., 2022*; *van der Schaaf et al., 2013*) and the mechanisms underlying this variability remain poorly understood.

There is evidence that the effects of catecholaminergic drugs depend on an individual's baseline dopamine levels in the prefrontal cortex and striatum (*Cohen and Servan-Schreiber, 1992*; *Cools and D'Esposito, 2011*; *Dodds et al., 2008*; *Durstewitz and Seamans, 2008*; *Rostami Kandroodi et al., 2021*; *van den Bosch et al., 2022*). Depending on individual baseline dopamine levels the administration of catecholaminergic drugs can promote states of cognitive flexibility or stability. For example, pushing dopamine from low to optimal (medium) levels may increase update thresholds in the light of new information (i.e., facilitating shielding/stability), whereas if a drug pushes dopamine either too high or too low may decrease update thresholds (i.e., facilitating shifting/flexibility) (*Durstewitz and Seamans, 2008*; *Goschke and Bolte, 2018*).

Here, we argue that baseline performance should be considered when studying the behavioral effects of catecholaminergic drugs effects. For example, *Rostami Kandroodi et al., 2021* reported that the re-uptake blocker methylphenidate did not alter reversal learning overall, but preferentially improved performance in participants with higher working memory capacity. To investigate the role of baseline performance in drug challenge studies, it is important to control for several factors. First, the order of drug and placebo (PL) sessions must be balanced to control for practice effects (*Bartels*

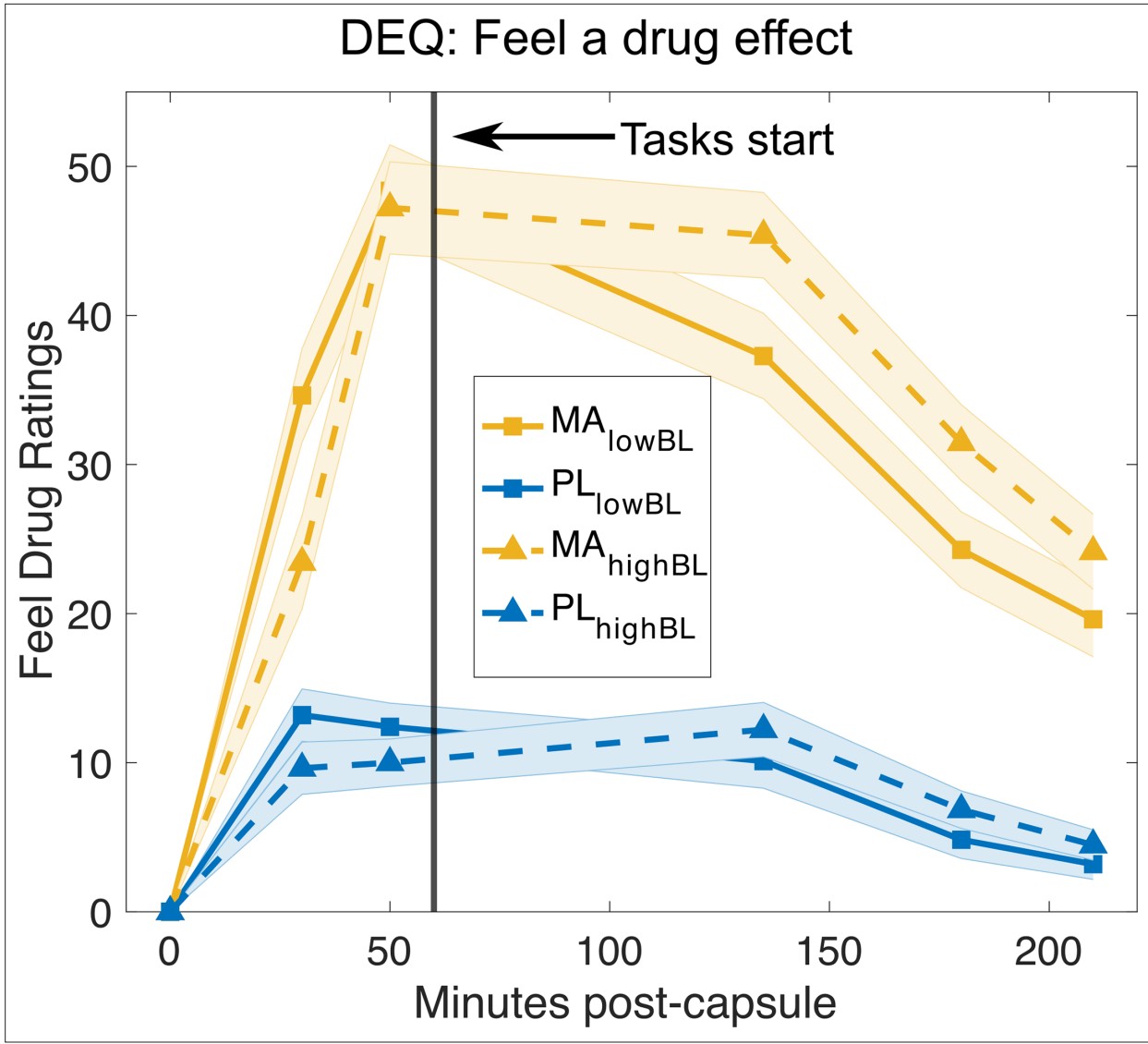

**Figure 1.** Subjective drug effects post-capsule administration. Methamphetamine (MA) increased 'feel drug effect' ratings compared to placebo. The scale for the ratings of Feeling a drug effect range from 0 to 100. The vertical black line indicates the time at which the task was started. Ratings of 'feeling' a drug effect did not differ significantly between low vs. high baseline performers (all p > 0.05). DEQ = Drug Effects Questionnaire (*Morean et al., 2013*). Mean/SEM = line/shading.

*et al., 2010*; *Garrett et al., 2015*; *MacRae et al., 1988*; *Servan-Schreiber et al., 1998*). Second, it is desirable to obtain an independent measure of baseline performance that is not confounded with the drug vs. PL comparison. Thus, participants may be stratified based on their performance on an independent session.

In the present study, we studied the effects of MA, a stimulant that increases monoaminergic transmission, on probabilistic reversal learning dynamics in a within-subject, double-blind, randomized design. The effects of the drug on a reversal learning task were examined in relation to participants' baseline level of performance. Baseline performance was determined during an initial drug-free session. Then, participants completed the task during two sessions after receiving PL and 20 mg of MA (order counterbalanced) (*Figure 1*).

The task used to study adaptive learning dynamics was a reversal variant of an established probabilistic learning task (*Fischer and Ullsperger, 2013*; *Jocham et al., 2014*; *Kirschner et al., 2022*; *Kirschner et al., 2024*). On each trial, subjects made a choice to either gamble or avoid gambling on a probabilistic outcome, in response to a stimulus presented in the middle of the screen (see

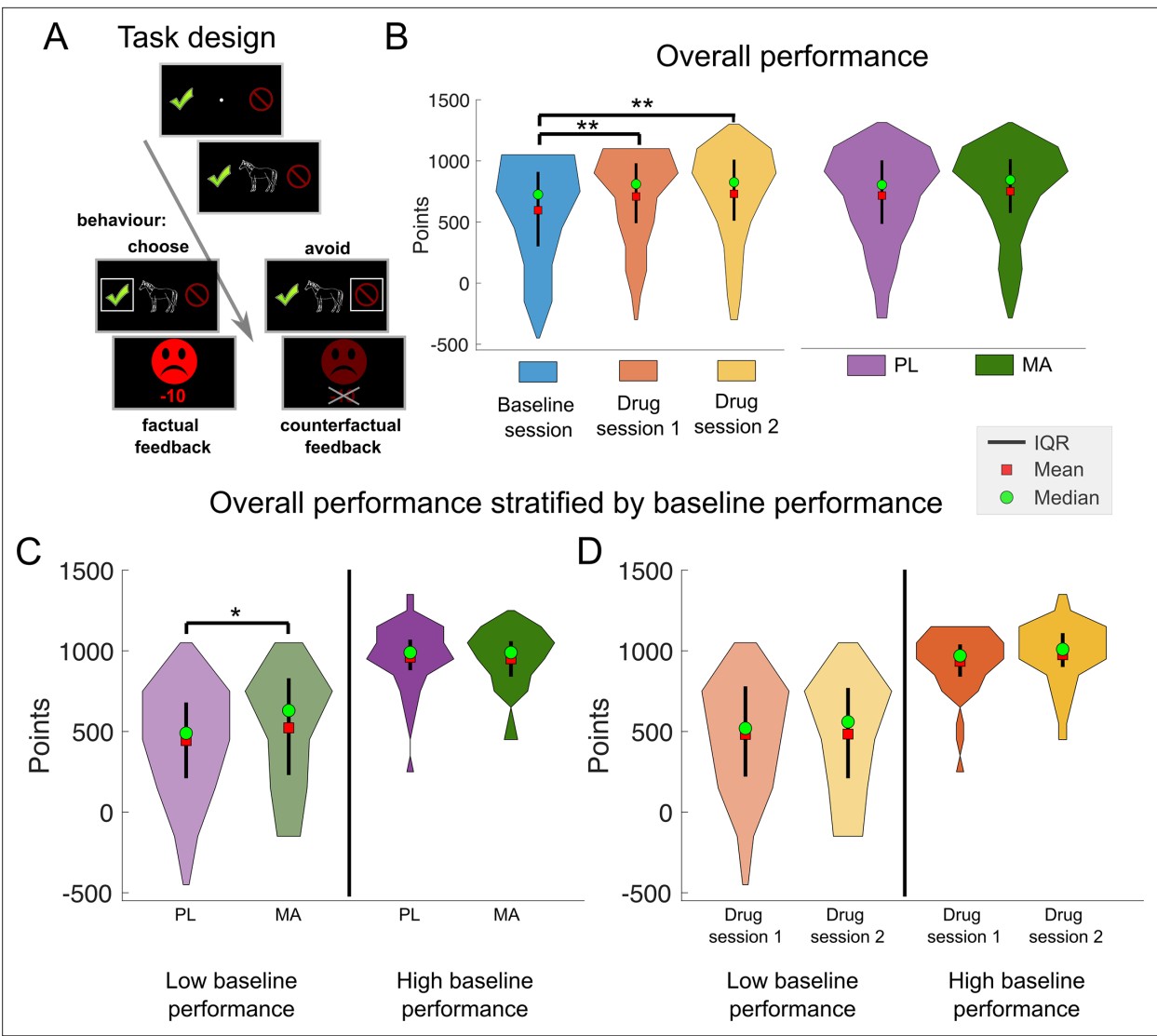

**Figure 2.** Methamphetamine improved performance in a modified probabilistic reversal learning task only in participants who performed the task poorly at baseline. (**A**) Schematic of the learning task. Each trial began with the presentation of a random jitter between 300 and 500 ms. Hereafter, a fixation cross was presented together with two response options (choose – green tick mark or avoid – red no-parking sign). After the fixation cross, the stimulus was shown centrally until the participant responded or for a maximum duration of 2000 ms. Thereafter, participants' choices were confirmed by a white rectangle surrounding the chosen option for 500 ms. Finally, the outcome was presented for 750 ms. If subjects chose to gamble on the presented stimuli, they received either a green smiling face and a reward of 10 points or a red frowning face and a loss of 10 points. When subjects avoided a symbol, they received the same feedback but with a slightly paler color and the points that could have been received were crossed out to indicate that the feedback was fictive and had no effect on the total score. A novel feature of this modified version of the task is that we introduced different levels of noise (probability) to the reward contingencies. Here, reward probabilities could be less predictable (30% or 70%), more certain (20% or 80%), or random (50%). (**B**) Total points earned in the task split up in sessions (baseline, drug sessions 1 and 2) and drug condition (PL vs. MA). Results show practice effects but no differences between the two drug sessions (baseline vs. drug session 1: 595.85 (39.81) vs. 708.62 (36.93); $t(93) = -4.21$, p = $5.95^{-05}$, $d = 0.30$; baseline vs. drug session 2: 595.85 (39.81) vs. 730.00 (38.53); $t(93) = -4.77$, p = $6.66^{-06}$, $d = 0.35$; session 1 vs. session 2: $t(93) = -0.85$, p = 0.399, $d = 0.05$). (**C**) Interestingly, when we stratified drug effects by baseline performance (using median split on total points at baseline), we found that there was a trend toward better performance under MA in the low baseline performance group ($n = 47$, p = 0.07). (**D**) Overall performance in drug sessions 1 and 2 stratified by baseline performance. Here, baseline performance appears not to affect performance in drug session 1 or 2. *Note.* IQR = inter quartile range; PL = placebo; MA = methamphetamine. ** = significant difference (p < .05); * = marginally significant difference (p = .07).

*Figure 2A*). A gamble could result in a gain or loss of 10 points, depending on the reward contingency associated with that stimulus. In choosing not to gamble, subjects avoided losing or winning points, but they were informed what would have happened if they had chosen to gamble. The reward contingency changed every 30–35 trials. By learning which symbols to choose and which to avoid,

participants could maximize total points. A novel feature of this modified version of the task is that we introduced different levels of noise (probability) to the reward contingencies. Here, reward probabilities could be less predictable (30% or 70%, i.e., high outcome noise) or more certain (20% or 80%; i.e., low outcome noise). This manipulation allowed us to study the effect of MA on the dynamic balancing of updating and shielding beliefs about reward contingencies within different levels of noise in the task environment. To estimate learning rate adjustments, we fit a nested set of reinforcement learning models, that allowed for trial-by-trial learning rate adjustments.

We found that MA improved participants' performance in the task, but this effect was driven mainly by a greater improvement in performance in those participants who performed poorly during the baseline session. Modeling results suggested that MA helps performance by adaptively shifting the relative weighting of surprising outcomes based on their statistical context. Specifically, MA facilitates down-weighting of probabilistic errors in stages of less predictable reward contingencies. Together, these results reveal novel insights into the role of catecholamines in adaptive learning behavior and highlights the importance to consider individual difference at baseline.

**Table 1.** Demographics and drug use characteristics of study participants ($n$ = 94).

| Demographic categories | Low baseline performers, $n$ (%) or mean (SD) | high baseline performers, $n$ (%) or mean (SD) |
|---|---|---|
| Sex (M/F) | 17/30 | 29/18 |
| | (36/64%) | (62/38%) |
| Age (years) | 24.3 (±3.9) | 24.7 (±4.1) |
| BMI | 23.1 (±2.6) | 23.3 (±2.3) |
| Education (years) | 15.8 (±1.6) | 15.7 (±1.6) |
| *Race/ethnicity* | | |
| American Indian/Alaskan | 1 (2%) | 0 (0%) |
| Asian | 10 (22%) | 8 (17%) |
| Black or African American | 2 (4%) | 0 (0%) |
| Hispanic | 7 (15%) | 6 (13%) |
| White | 24 (51%) | 26 (55%) |
| More than one race | 1 (2%) | 6 (13%) |
| Not reported | 2 (4%) | 1 (2%) |
| *Drug use (past month)* | | |
| Caffeinated drinks/day | 1.0 (±0.8) | 1.3 (±1.2) |
| Alcoholic drinks/week | 3.8 (±3.0) | 3.9 (±2.6) |
| Cannabis uses/month | 4.2 (±6.7) | 2.5 (±4.6) |
| Daily nicotine users | 2 (4%) | 1 (2%) |
| *Lifetime stimulant use* | | |
| People who previously used at least once (prescription) | 3 (6%) | 2 (4%) |
| People who previously used at least once (recreationally) | 8 (17%) | 9 (19%) |
| *Other lifetime drug use (median number of times used)* | | |
| Cannabis | 30.5 | 20 |
| Opiates | 0 | 0 |
| Hallucinogens | 1 | 0 |

## Results

Ninety-four healthy young adults completed the probabilistic learning task (*Figure 2*; *Fischer and Ullsperger, 2013*; *Jocham et al., 2014*; *Kirschner et al., 2022*; *Kirschner et al., 2024*) on three separate sessions, an initial drug-free session, and after PL and MA. The study followed a double-blinded cross-over design, whereby 50% of participants received MA first, and 50% of participants PL first. *Table 1* shows the demographic characteristics of the participants grouped by their task performance during the baseline session. The groups did not differ significantly on any of the variables measured. In a first analysis, we checked for general practice effects across the three task completions based on the total points earned in the task. We found a strong practice effect ($F(2,186) = 14.53$, p < 0.001) with better performance on session two and three compared to session one (baseline). There was no difference in the total scores between session two and three (see *Figure 2B*). These results suggest that the baseline session may have minimized order effects between MA and PL sessions (see also results and discussion). The key findings detailed below are summarized in a schematic figure presented in the discussion section (Figure 10).

## Subjective drug effects

MA administration significantly increased 'feel drug effect' ratings compared to PL, at 30, 50, 135, 180, and 210 min post-capsule administration (see *Figure 1*; Drug × Time interaction $F(5,555) = 38.46$, p < 0.001). In the MA session, no differences in the 'feel drug effect' were observed between low and high baseline performer, including peak change-from-baseline ratings (rating at 50 min post-capsule: low baseline performer: 48.36 (4.29) vs. high baseline performer: 47.21 (4.44); $t(91) = 0.18$, p = 0.85, $d = 0.03$; rating at 135 min post-capsule: low baseline performer: 37.27 (4.15) vs. high baseline performer: 45.38 (3.84); $t(91) = 1.42$, p = 0.15, $d = 0.29$).

### Drug effects on overall performance and RT

In general, participants learned the task well, based on the observation that their choice behavior largely followed the underlying reward probabilities of the stimuli across the sessions (see Figure 4D-F). When all subjects were considered together, we did not find a performance benefit under MA quantified by the total points scored in the task (MA: 736.59 (37.11) vs. PL: 702.02 (38.305); $t(93) = 1.38$, p = 0.17, $d = 0.10$). When participants were stratified by their baseline performance (median spilt on total points at baseline), we found a marginally significant Drug × Baseline Performance Group interaction (Drug × Baseline Performance Group interaction: $F(1,92) = 3.20$, p = 0.07; see *Figure 2C* and 7A). Post hoc *t*-tests revealed that compared to PL, MA improved performance marginally in participants with poor baseline performance (total points MA: 522.55 (53.79) vs. PL: 443.61 (47.81); $t(46) = 1.85$, p = 0.071, $d = 0.23$). MA did not, however, improve performance in the high baseline performance group (total points MA: 950.63 (26.15) vs. PL: 960.42 (27.26); $t(46) = –0.38$, p = 0.698, $d = 0.05$). In control analyses, we ensured that these effects are not driven by session-order effects (see also section on session control analyses). Results showed no effect of Session ($F(1,92) = 0.71$, p = 0.40) and no Session × Baseline Performance Group interaction ($F(1,92) = 0.59$, p = 0.44; see *Figure 1C*). There was a trend for slightly faster RTs under MA (PL: 544.67 ms (9.87) vs. MA: 533.84 ms (11.51); $t(93) = 1.75$, p = 0.08, $d = 0.10$). This speed effect appeared to be independent of baseline performance (Drug × Baseline Performance Group interaction: $F(1,92) = 0.45$, p = 0.50). Moreover, MA was associated with reduced RT variability (average individual SD of RTs: PL: 193.74 (6.44) vs. MA: 178.98 (5.47); $t(93) = 2.54$, p = 0.012, $d = 0.25$). Reduced RT variability has previously been associated with increased attention and performance (*Esterman et al., 2013*; *Karamacoska et al., 2018*). Two-way ANOVA on RT variability revealed an effect of baseline performance ($F(1,92) = 4.52$, p = 0.03), with increased RT variability in low baseline performers across the drug sessions (low baseline performance: 197.27 (6.48) vs. high baseline performance: 175.45 (5.29)). Moreover, there was an effect of Drug ($F(1,92) = 6.87$, p = 0.01), and a Drug × Baseline Performance Group interaction ($F(1,92) = 6.97$, p = 0.009). Post hoc *t*-tests indicated that the MA-related reduction in RT variability was specific to low baseline performers (PL: 212.07 (9.84) vs. MA: 182.46 (7.98); $t(46) = 3.04$, p = 0.003, $d = 0.48$), whereas MA did not affect high baseline performers RT variability (PL: 175.40 (7.51) vs. MA: 175.50 (7.55); $t(46) = –0.02$, p = 0.98, $d < 0.01$).

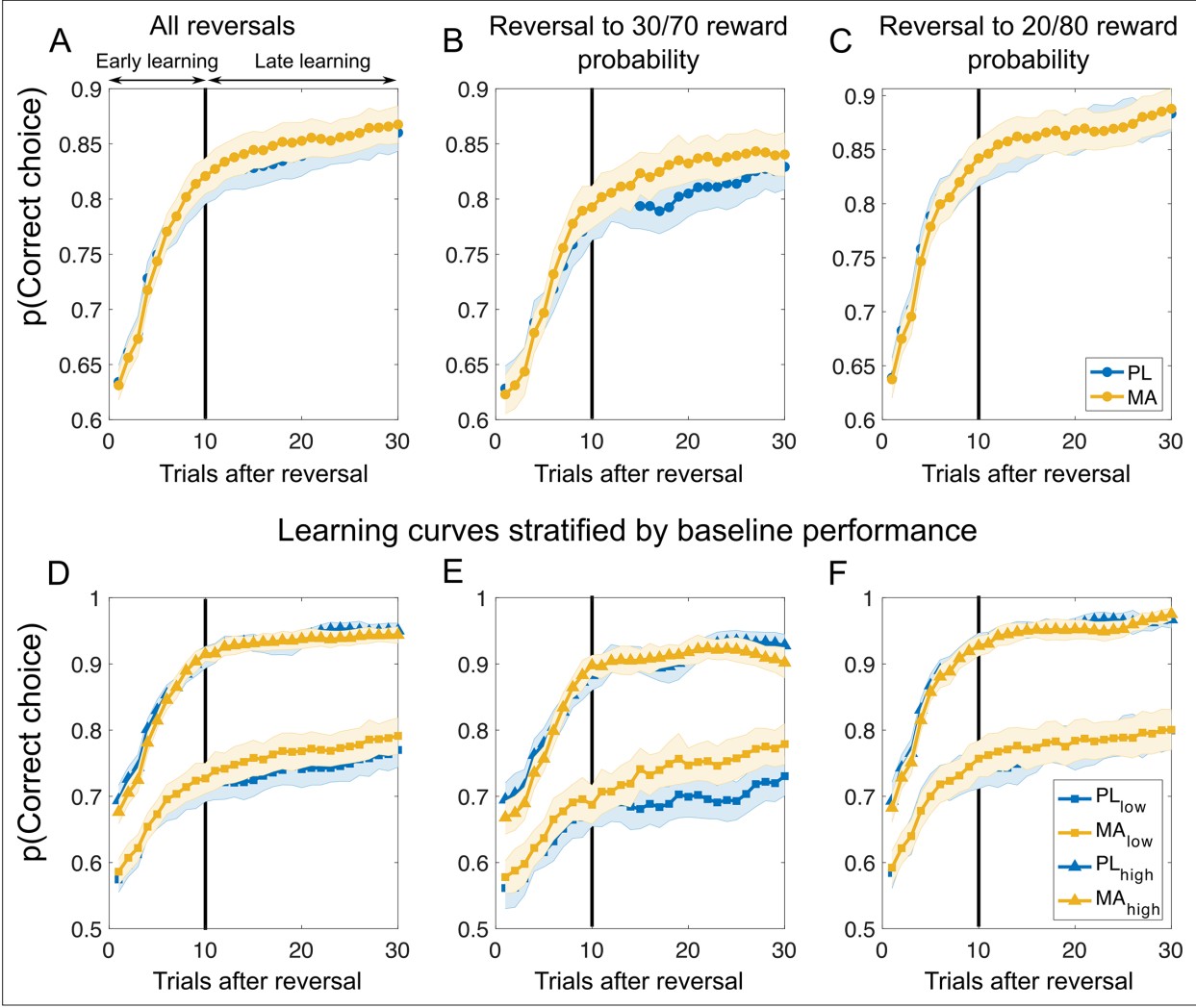

**Figure 3.** Learning curves after reversals suggest that methamphetamine improves learning performance in phases of less predictable reward contingencies in low baseline performer. Top panel of the figure shows learning curves after all reversals (**A**), reversals to stimuli with less predictable reward contingencies (**B**), and reversals to stimuli with high reward probability certainty (**C**). Bottom panel displays the learning curves stratified by baseline performance for all reversals (**D**), reversals to stimuli with less predictable reward probabilities (**E**), and reversals to stimuli with high reward probability certainty (**F**). Vertical black lines divide learning into early and late stages as suggested by the Bai–Perron multiple break point test. Results suggest no clear differences in the initial learning between MA and PL. However, learning curves diverged later in the learning, particular for stimuli with less predictable rewards (**B**) and in subjects with low baseline performance (**E**). *Note*. PL = placebo; MA = methamphetamine; Mean/SEM = line/shading. Data plots are smoothed with a running average (±2 trials), leading to slightly overestimated values after reversals. Additional analyses confirm that the probability of choosing the correct response after reversals is not above chance level (*t*-test against chance: all reversals: $t(93) = 1.64$, $p = 0.10$, $d = 0.17$, 99% CI [0.49,0.55]; reversal to low outcome noise: $t(93) = 1.67$, $p = 0.10$, $d = 0.17$, 99% CI [0.49,0.56]; reversal to high outcome noise: $t(93) = 0.87$, $p = 0.38$, $d = 0.09$, 99% CI [0.47,0.56]).

The online version of this article includes the following figure supplement(s) for figure 3:

**Figure supplement 1.** Learning curves for different reversal types.

## MA improves learning performance when reward contingencies are less predictable

Next, to get a better understanding of how MA affects learning dynamics, we investigated the probability of correct choice (i.e., choosing the advantageous stimuli and avoiding disadvantageous stimuli) across successive reversals. As shown in *Figure 3*, the drug did not affect initial learning. However, the drug improved performance later in learning, particularly for stimuli with less predictable reward probabilities (see *Figure 3B*) and in subjects with low baseline performance. To quantify this observation, we first applied the Bai–Perron multiple break point test (see Methods) to find systematic breaks

in the learning curves allowing us to divide learning into early and late stages. We applied the test to the reversal learning data across subjects. One break point was identified at 10 trials after a reversal (indexed by the vertical lines in *Figure 3*). We did not find drug differences when considering all reversals (PL: 0.84 (0.01) vs. MA: 0.85 (0.01); $t(93) = -1.14$, p = 0.25, $d = 0.07$) and reversals to stimuli with high reward probability certainty (PL: 0.86 (0.01) vs. MA: 0.87 (0.01); $t(93) = -0.25$, p = 0.80, $d = 0.02$). Interestingly, we found a trend for increased learning under MA for stimuli with less predictable rewards (PL: 0.80 (0.01) vs. MA: 0.82 (0.01); $t(93) = -1.80$, p = 0.07, $d = 0.14$). Two-way ANOVA on the averaged probability of correct choice during the late stage of learning revealed a Drug × Baseline Performance Group interaction ($F(1,92) = 4.85$, p = 0.03; see Figure 7B). Post hoc $t$-tests revealed that subjects performing lower at baseline appeared to benefit from MA (average accuracy late learning PL: 0.69 (0.02) vs. MA: 0.74 (0.02); $t(46) = -2.59$, p = 0.01, $d = 0.32$), whereas there was no difference between MA and PL in the high baseline performance group (PL: 0.91 (0.01) vs. MA: 0.91 (0.01); $t(46) = 0.29$, p = 0.77, $d = 0.04$). We did not find other differences in reversal learning (all p > 0.1). In control analyses we split the learning curves into other possible learning situations in the task (i.e., acquisition, first reversal learning, etc.). Here, no drug-related effects emerged (see *Figure 3—figure supplement 1*).

## Computational modeling results

To gain a better mechanistical understanding of the trial-to-trial learning dynamics we constructed a nested model set built from RL models (see methods) that included the following features: (1) an inverse temperature parameter of the softmax function used to convert trial expected values to action probabilities ($\beta$), (2) a play bias term that indicates a tendency to attribute higher value to gambling behavior, and (3) an intercept term for the effect of learning rate on choice behavior. Additional parameters controlled trial-by-trial modulations of the learning rate including feedback confirmation (confirmatory feedback was defined as factual wins and counterfactual losses, disconfirmatory feedback was defined as factual losses and counterfactual wins), feedback modality (factual vs. counterfactual) and weighting of the learning rate as a function of the absolute value of previous prediction error (parameter Eta, determining the influence of surprise about the outcome on learning; *Li et al., 2011*). The winning model (as measured by lowest Bayesian information criterion [BIC] and achieving protected exceedance probabilities of 100%) was one that allowed the learning rate to vary based on whether the feedback was confirmatory or not and the level of surprise of the outcome (see *Figure 4A*). Sufficiency of the model was evaluated through posterior predictive checks that matched behavioral choice data (see *Figures 4D–F and 5*) and model validation analyses (see *Figure 4—figure supplement 1*). Specifically, using individual maximum likelihood parameter estimates, we simulated task performance and confirmed that MA increases performance later in learning for stimuli with less predictable reward probabilities, particularly in subjects with low baseline performance (*Figure 5A*; mean ± SD: simPL low performance: 0.69 ± 0.01 vs. simMA low performance: 0.72 ± 0.01; $t(46) = 2.00$, p = 0.03, $d = 0.23$). We did not find evidence for differences in model fit between the groups (avg. BIC PL: 596.77 (21.63) vs. MA: 599.66 (19.85); $t(93) = -0.25$, p = 0.80, $d = 0.01$).

Next, we compared MAs effect on best-fitting parameters of the winning model (see *Figure 4B, C*). We found that eta (the parameter controlling dynamic adjustments of learning rate according to recent absolute prediction errors) was reduced under MA (eta MA: 0.24 (0.01) vs. PL: 0.30 (0.01); $t(93) = -3.005$, p = 0.003, $d = 0.43$). When we stratified drug effects by baseline performance, we found a marginally significant Drug × Baseline Performance Group interaction ($F(1,92) = 3.09$, p = 0.08; see Figure 7C). Post hoc t tests revealed that compared to PL, MA affected eta depending on baseline performance in the task. Here, subjects performing less well at baseline showed smaller eta's (eta MA: 0.24 (0.01) vs. 0.33 (0.02); $t(46) = -3.06$, p = 0.003, $d = 0.67$), whereas there was no difference between MA and PL in the high baseline performance group MA: 0.23 (0.01) vs. 0.26 (0.01); $t(46) = -1.03$, p = 0.31, $d = 0.18$. We did not find drug-related differences in any other model parameters (all p > 0.1).

## MA affects learning rate dynamics

Next, we investigated how the model parameters fit with trial-by-trial modulations of the learning rate. Learning rates in our best-fitting model were dynamic and affected by both model parameters and their interaction with feedback. Learning rate trajectories after reversals are depicted in *Figure 6*. As suggested by lower eta scores, MA appears to be associated with reduced learning rate dynamics

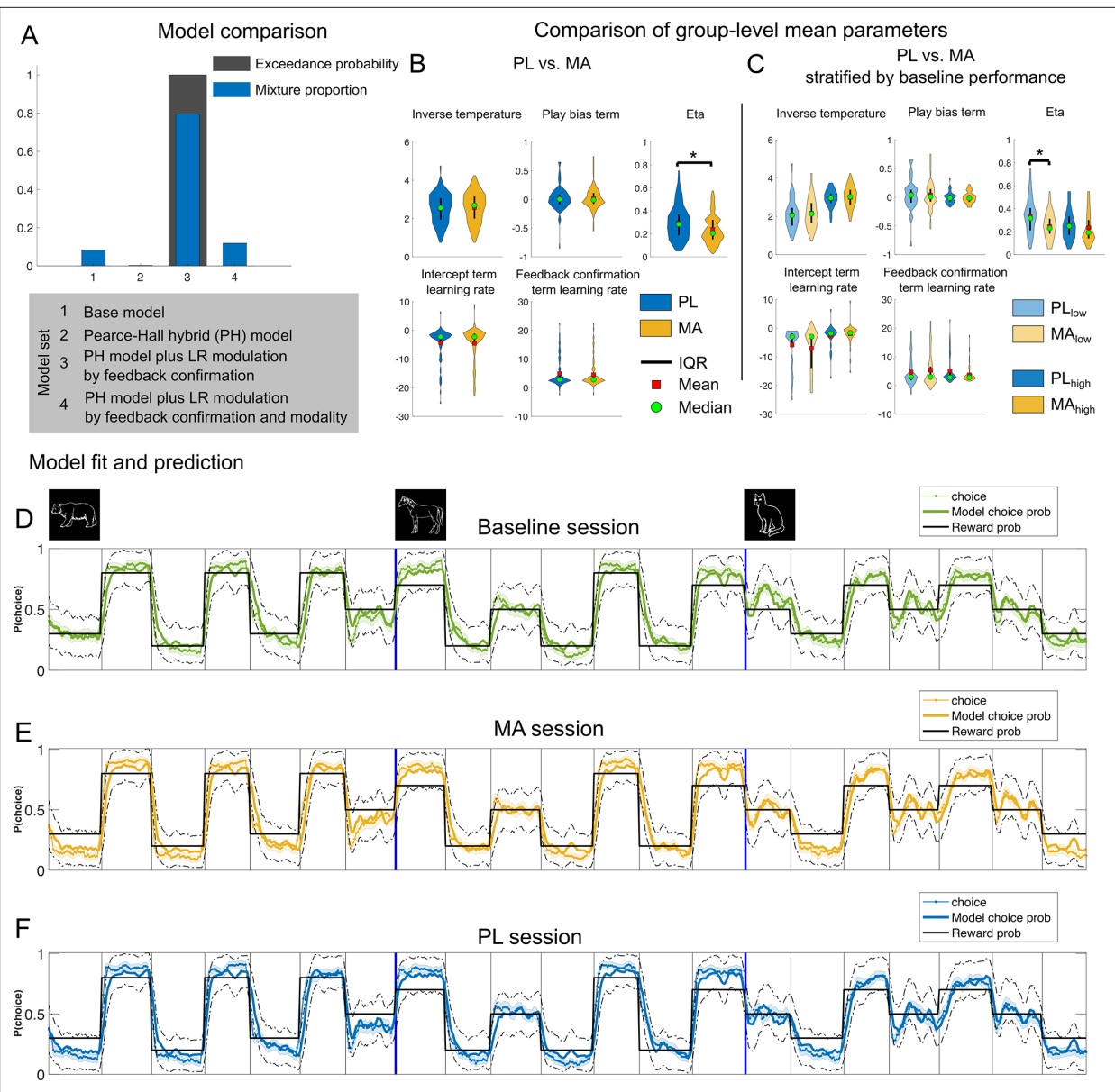

**Figure 4.** Computational modeling results reveal that methamphetamine affects the model parameter controlling dynamic adjustments of learning rate. (**A**) Model comparison. Bayesian model selection was performed using –0.5*BIC as a proxy for model evidence (***Stephan et al., 2009***). The best-fitting mixture model assigned proportions to each model based on the frequency with which they provided the 'best' fit to the observed participant data (Mixture proportion; blue bars) and estimated the probability with which the true population mixture proportion for a given model exceeded that of all others (Exceedance probability; black bars). The hybrid model plus learning rate modulation by feedback confirmatory (model 3) provided the best fit to the majority of participants and had an exceedance probability near one in our model set. (**B, C**) Comparison of parameter estimates from the winning model on-/off-drug. Stars indicate significant difference for the respective parameter. Results suggest that only the parameter controlling dynamic adjustments of learning rate according to recent prediction errors, eta, was affected by our pharmacological manipulation. (**D–F**) Modeled and choice behavior of the participants in the task, stretched out for all stimuli. Note that in the task the different animal stimuli were presented in an intermixed and randomized fashion, but this visualization allows to see that participants' choices followed the reward probabilities of the stimuli. Data plots are smoothed with a running average (±2 trials). Ground truth corresponds to the reward probability of the respective stimuli (good: 70/80%; neutral: 50%; bad: 20/30%). Dashed black lines represent 95% confidence intervals derived from 1000 simulated agents with parameters that were best fit to participants in each group. Model predictions appear to capture the transitions in choice behavior well. Mean/SEM = line/shading. *Note.* IQR = inter quartile range; PL = placebo; MA = methamphetamine.

The online version of this article includes the following figure supplement(s) for figure 4:

**Figure supplement 1.** Validation of model selection and parameter recovery.

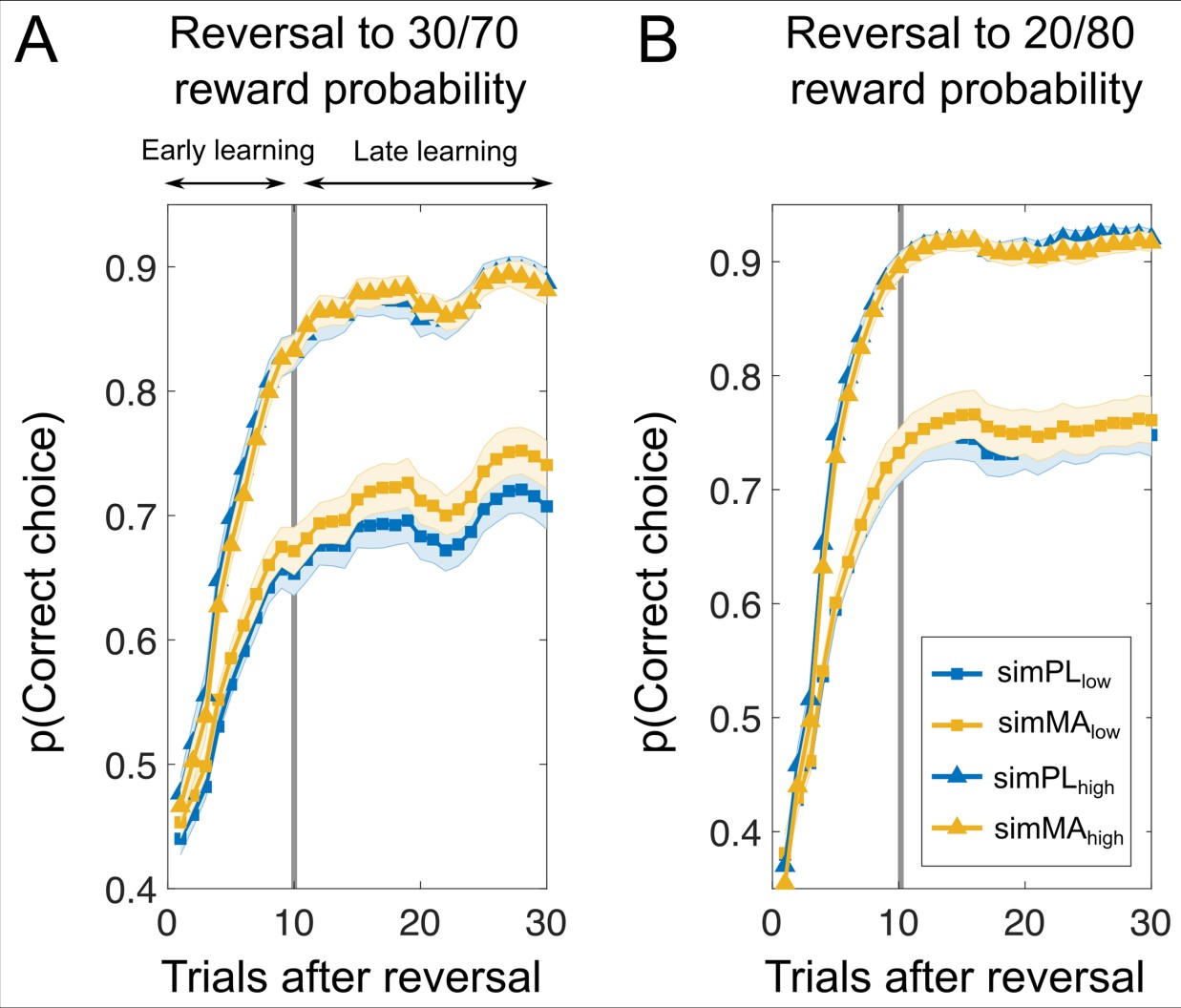

**Figure 5.** Simulated task performance based on individual maximum likelihood parameter estimates reflects drug-induced behavioral differences. Simulated performance (*y*-axis) is plotted against trials after reversals (*x*-axis) for low (blue) and high (yellow) baseline performers. Using each participant's estimated parameters, 100 artificial agents were simulated playing the task, and their choices were averaged to represent each participant's behavior. The simulation shows that methamphetamine (MA) increases performance later in learning for stimuli with high outcome noise, particularly in subjects with low baseline performance (**A**). In contrast, no drug effect was observed for stimuli with low outcome noise (**B**). Note: Mean/SEM = line/shading.

in low baseline performers. In contrast, low baseline performers in the PL condition exhibited greater variability in learning rate (and average LR throughout) rendering their choices more erratic. Consistent with this, on many trials their choices were driven by the most recent feedback, as their learning rates on a large subset of trials in later learning stages (on average 9 out of 11; *Figure 6H*) were greater than 0.5. Specifically, variability in learning rate (average individual SD of learning rate) was reduced in both early and late stages of learning across all reversals (early PL: 0.20 (0.01) vs. MA: 0.17 (0.01); *t*(93) = 2.72, p = 0.007, *d* = 0.36; late PL: 0.18 (0.01) vs. MA: 0.15 (0.01); *t*(93) = 2.51, p = 0.01, *d* = 0.33), as were reversals to stimuli with less predictable rewards (early PL: 0.19 (0.01) vs. 0.16 (0.01); *t*(93) = 2.98, p = 0.003, *d* = 0.39; late PL: 0.18 (0.01) vs. MA: 0.16 (0.01); *t*(93) = 2.66, p = 0.009, *d*=0.35). Reversals to stimuli with high outcome certainty were also associated with decreased learning rate variability after MA administration (early PL: 0.18 (0.01) vs. MA: 0.15 (0.01); *t*(93) = 2.57, p = 0.01, *d* = 0.34; late PL: 0.18 (0.01) vs. MA: 0.15 (0.01); *t*(93) = 2.63, p = 0.009, *d* = 0.35). Two-way ANOVA revealed that this effect depended on baseline performance across all reversals (Drug × Baseline performance: *F*(1,92) = 3.47, p = 0.06), reversals to stimuli with less predictable rewards (Drug × Baseline performance: *F*(1,92) = 4.97, p = 0.02), and stimuli with high outcome certainty (Drug ×

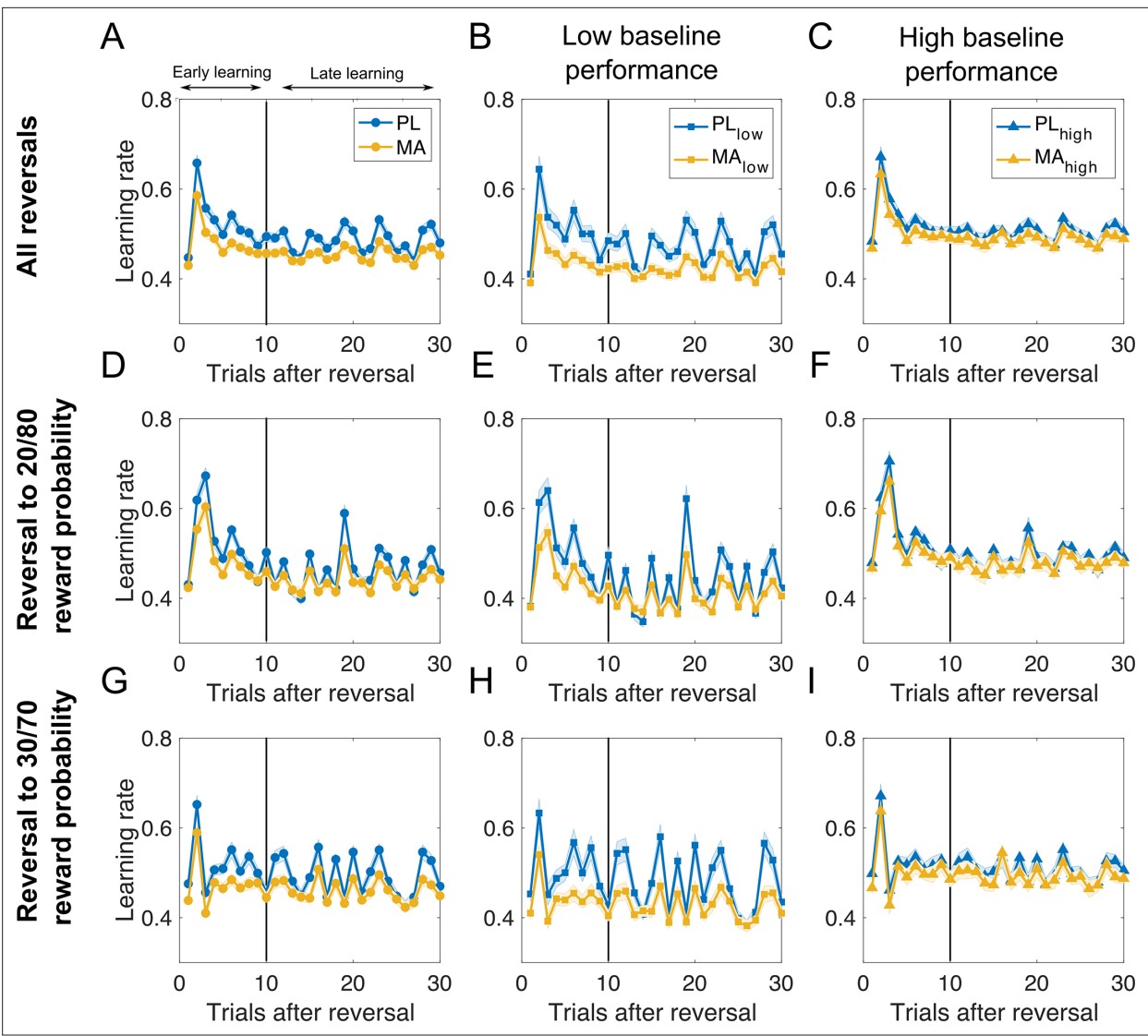

**Figure 6.** Methamphetamine boosts signal-to-noise ratio (SNR) between real reversals and misleading feedback in late learning stages. Learning rate trajectories after reversal derived from the computational model. First column depicts learning rates across all subjects for all reversals (**A**), reversal to stimuli with high reward probability certainty (**D**), and reversal to stimuli with noisy outcomes (**G**). Middle and right column shows learning rate trajectories for subjects stratified by baseline performance (**B, E, H** – low baseline performance; **C, F, I** – high baseline performance). Results suggest that people with high baseline performance show a large difference in learning rates after true reversals and during the rest of the task including misleading feedback. Specifically, they show a peak in learning after reversals and reduced learning rates in later periods of a learning block, when choice preferences should ideally be stabilized (**C**). This results in a better SNR between real reversals and misleading feedback (i.e., surprising outcomes in the late learning stage). In low baseline performers the SNR is improved after the administration of MA. This effect was particularly visible in stages of the task where rewards were less predictable (**H**). *Note.* PL = placebo; MA = methamphetamine; Mean/SEM = line/shading.

Baseline performance: $F(1,92) = 5.26$, $p = 0.03$). Here, reduced variability under MA was observed in low baseline performers (all $p < 0.006$, all $d > 0.51$) but not in high baseline performers (all $p > 0.1$). Together, these patterns of results suggest that people with high baseline performance show a large difference in learning rates after true reversals and during the rest of the task including misleading feedback. Specifically, they show a peak in learning after reversals and reduced learning rates in later periods of a learning block, when choice preferences should ideally be stabilized (see *Figure 6C*). This results in a better signal-to-noise ratio (SNR) between real reversals and misleading feedback (i.e., surprising outcomes in the late learning stage). In low baseline performers the SNR is improved after the administration of MA. This effect was particularly visible in stages of the task where rewards were less predictable. To quantify the SNR for less predictable reward contingencies for low baseline

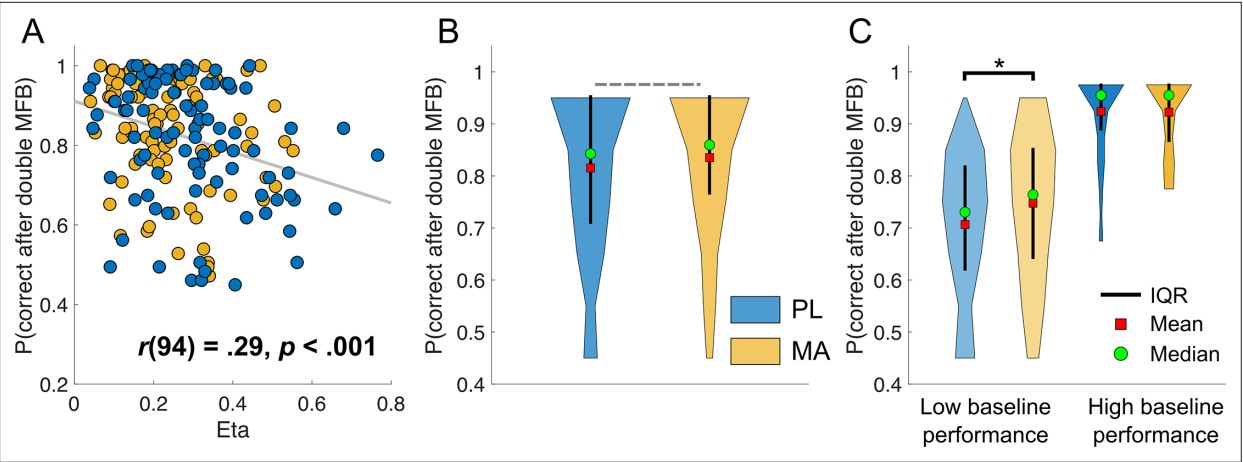

**Figure 7.** Misleading feedback effects on choice accuracy are modulated by eta and methamphetamine in low baseline performers. This figure shows the association between receiving misleading feedback later in learning (i.e., reward or losses that do not align with a stimulus' underlying reward probability) and the probability of making the correct choice during the next encounter of the same stimulus. Results indicate a negative correlation between the probability of a correct choice after double-misleading feedback and eta (**A**). Here, the probability of a correct choice after double-misleading feedback decreases with increasing eta. There was a trend (p = 0.06) that subjects under MA were more likely to make the correct choice after two misleading feedback as compared to PL (**B**). (**C**) This effect appeared to be dependent on baseline performance, whereby only subjects with low baseline performance seem to benefit from MA (p = 0.02). *Note.* IQR = inter quartile range; PL = placebo; MA = methamphetamine; MFB = misleading feedback; * = significant difference (p < .05).

performers, we computed the difference between learning rate peaks on true reversals (signal) vs. learning rate peaks after probabilistic feedback later in learning (noise; SNR). The results of this analysis revealed that MA significantly increased the SNR for low baseline performers (PL: 0.01 (0.01) vs. MA: 0.04 (0.01); $t(46) = -2.81$, $p = 0.007$, $d = 0.49$). Moreover, learning rates were generally higher in later stages of learning, when choice preferences should ideally have stabilized (avg. learning rate during late learning for less predictable rewards: PL: 0.48 (0.01) vs. MA: 0.42 (0.01); $t(46) = 3.36$, $p = 0.001$, $d = 0.56$).

Thus far, our results suggest that (1) MA improved performance in subjects who performed poorly at baseline, and (2) MA reduced learning rate variability in subjects with low baseline performance (driven by significantly lower eta parameter estimates, which improved the SNR between true reversals and misleading feedback particularly for less predictable rewards). Next, we aimed to test how these differences relate to each other. Given that eta causes increased learning after surprising feedback and that we found the biggest drug differences in later stages of learning for stimuli that have less predictable rewards, we tested the association between the probability of making the correct choice after two consecutive probabilistic errors (wins for bad stimuli and losses for good stimuli; in total this happened eight times in the late learning stage for stimuli with 30/70% reward probability) and eta. We found a significant correlation across participants (see **Figure 7**), whereby higher etas scores were associated with fewer correct choices ($r = 0.29$, $p = <0.001$). There was a trend toward a drug effect, with subjects in MA condition being more likely to make the correct choice after two misleading feedbacks (PL: 0.82 (0.02) vs. MA: 0.84 (0.01); $t(93) = -1.92$, $p = 0.06$, $d = 0.13$). Two-way ANOVA revealed, that this effect depended on baseline performance (Drug × Baseline performance: $F(1,92) = 4.27$, $p = 0.04$). Post hoc $t$-tests indicated higher correct choice probabilities under MA in low baseline performers (PL: 0.70 (0.02) vs. MA: 0.75 (0.02); $t(46) = -2.41$, $p = 0.02$, $d = 0.30$) but not in high baseline performers (PL: 0.92 (0.01234) vs. MA: 0.92 (0.01); $t(46) = 0.11$, $p = 0.91$, $d = 0.01$).

## MA shifts learning rate dynamics closer to the optimum for low baseline performers

To better understand the computational mechanism through which MA improved performance in low baseline performers, we first examined how performance in the task related to model parameters from our fits. To do so, we regressed task performance onto an explanatory matrix containing model parameter estimates across all conditions (see **Figure 8A**). The results of this analysis revealed

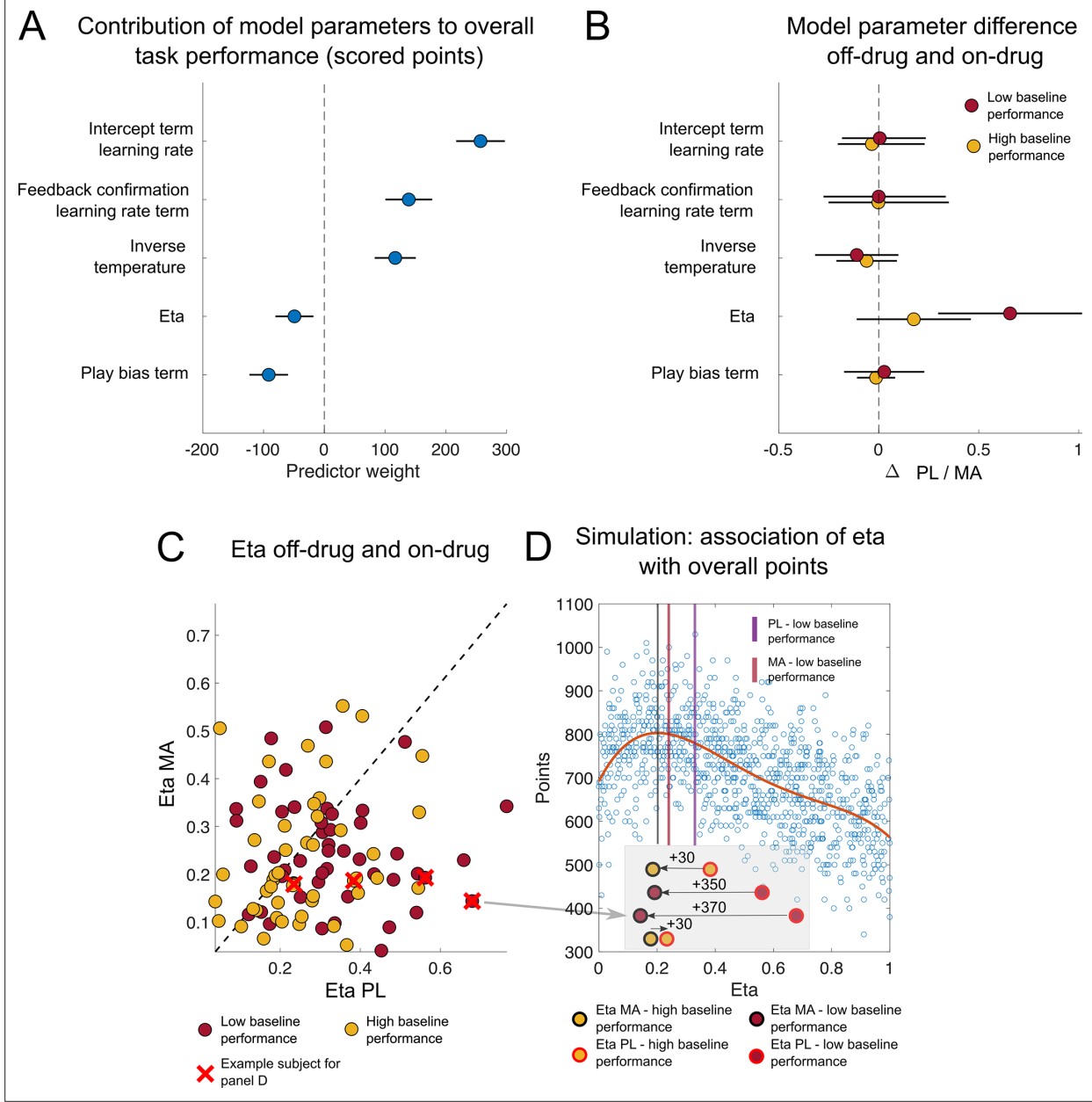

**Figure 8.** Changes in learning rate adjustment explain drug-induced performance benefits in low baseline performers. (**A**) Regression coefficients and 95% confidence intervals (points and lines; sorted by value) stipulating the contribution of each model parameter estimate to overall participants task performance (i.e., scored points in the task). Play bias and eta (the parameter governing the influence of surprise on learning rate) both made a significant negative contribution to overall task performance, whereas inverse temperature and learning rates were positively related to performance. (**B**) Differences in parameter values for on- and off-drug sessions as quantified by regression coefficients and 95% confidence intervals are plotted separately for high (red) and low (yellow) baseline performers. Note that the drug predominately affected the eta parameter and did so to a greater extent in low baseline performers. (**C**) eta estimates on-drug (*y*-axis) are plotted against eta estimates off-drug (*x*-axis) for high baseline performer (yellow points) and low baseline performer (red points). Note that a majority of subjects showed a reduction in eta on- vs. off-drug (67.02%). This effect was more pronounced in low baseline performers (low baseline performers: 74.47%; low baseline performers: 59.57%). (**D**) To better understand how changes in eta might have affected overall performance we conducted a set of simulations using the parameters best fit to human subjects, except that we equipped the model with a range of randomly chosen eta values to examine how altering that parameter might affect performance (*n* = 1000 agents). The results revealed that simulated agents with low-to-intermediate levels of eta achieved the best task performance, with models equipped with the highest etas performing particularly poorly. To illustrate how this relationship between eta and performance could have driven improved performance for some participants under the methamphetamine condition, we highlight four participants with low–moderate eta values under methamphetamine, but who differ dramatically in their eta values in the placebo condition (**D**, inset). PL = placebo; MA = methamphetamine.

that variability in several of the parameters was related to overall task performance, with the overall learning rate, feedback confirmation LR adjustments, and inverse temperature all positively predicting performance and eta and the play bias term negatively predicting it.

While each of these parameters explained unique variance in overall performance levels, only the parameter controlling dynamic adjustments of learning rate according to recent prediction errors, eta, was affected by our pharmacological manipulation (*Figure 6B*). In particular, eta was reduced in the MA condition, specifically in the low baseline group, albeit to an extent that differed across individuals (*Figure 8C*). To better understand how changes in eta might have affected overall performance we conducted a set of simulations using the parameters best fit to human subjects, except that we equipped the model with a range of randomly chosen eta values, to examine how altering that parameter might affect performance. The results revealed that simulated agents with low to intermediate levels of eta achieved the best task performance, with models equipped with the highest etas performing particularly poorly (*Figure 8D*). To illustrate how this relationship between eta and performance could have driven improved performance for some participants under the MA condition, we highlight four participants with low–moderate eta values under MA, but who differ dramatically in their eta values in the PL condition (*Figure 8D*, inset). Note that the participants who have the largest decreases in eta under the MAs, resulting from the highest PL levels of eta, would be expected to have the largest improvements in performance. It is noteworthy that low baseline performers tended to have particularly high values of eta under the baseline condition (low baseline performers: 0.33 (0.02) vs. high baseline performers: 0.25 (0.01); $t(46) = 2.59$, $p = 0.01$, $d = 0.53$), explaining why these individuals saw the largest improvements under the MA condition. Taken together, these results suggest that MA alters performance by changing the degree to which learning rates are adjusted according to recent prediction errors (eta), in particular by reducing the strength of such adjustments in low baseline performers to push them closer to task-specific optimal values.

While eta seemed to account for the differences in the effects of MA on performance in our low and high performance groups, it did not fully account for performance differences across the two groups (see *Figure 1C* and 10A, B). When comparing other model parameters between low and high baseline performer across drug sessions, we found that high baseline performer displayed higher overall inverse temperatures (2.97 (0.05) vs. 2.11 (0.08); $t(93) = 7.94$, $p < 0.001$, $d = 1.33$). This suggests that high baseline performers displayed higher transfer of stimulus values to actions leading to better performance (as also indicated by the positive contribution of this parameter to overall performance in the GLM). Moreover, they tended to show a reduced play bias (–0.01 (0.01) vs. 0.04 (0.03); $t(93) = -1.77$, $p = 0.08$, $d = 0.26$) and increased intercepts in their learning rate term (–2.38 (0.364) to –6.48 (0.70); $t(93) = 5.03$, $p < 0.001$, $d = 0.76$). Both of these parameters have been associated with overall performance (see *Figure 8A*). Thus, overall performance difference between high and low baseline performed can be attributed to differences in model parameters other than eta. However, as described in the previous paragraph, differential effects of MA on performance on the two groups were driven by eta.

This pattern of results suggests that MA specifically affects the eta parameter while leaving other parameters, such as the inverse temperature, unaffected. This points to a selective influence on a single computational mechanism. To verify this conclusion, we extended the winning model by allowing each parameter, in turn, to be differentially estimated for MA and PL, while keeping the other parameters fixed at the group (low and high baseline performance) mean estimates of the winning model for the PL session. These control analyses confirmed that MA affects only the eta parameter in the low-performer group (see *Appendix 3—table 1*) and that parameters did not tradeoff in our model. A similar effect was observed in a previous study investigating the effects of catecholaminergic drug administration on a probabilistic reversal learning task (*Rostami Kandroodi et al., 2021*). In that study, methylphenidate was shown to influence the inverse learning rate parameter as a function of working memory span, assessed through a baseline cognitive task. Consistent with our findings, no drug effects were observed on other parameters in their model, including the inverse temperature.

## MA may reduce misinterpretation of high outcome noise in low performer

In our task, outcomes are influenced by two distinct sources of noise: process noise (volatility) and outcome noise (stochasticity). Optimal learning rate should increase with volatility and decrease with stochasticity. Volatility was fairly constant in our task (change points around every 30–35 trials).

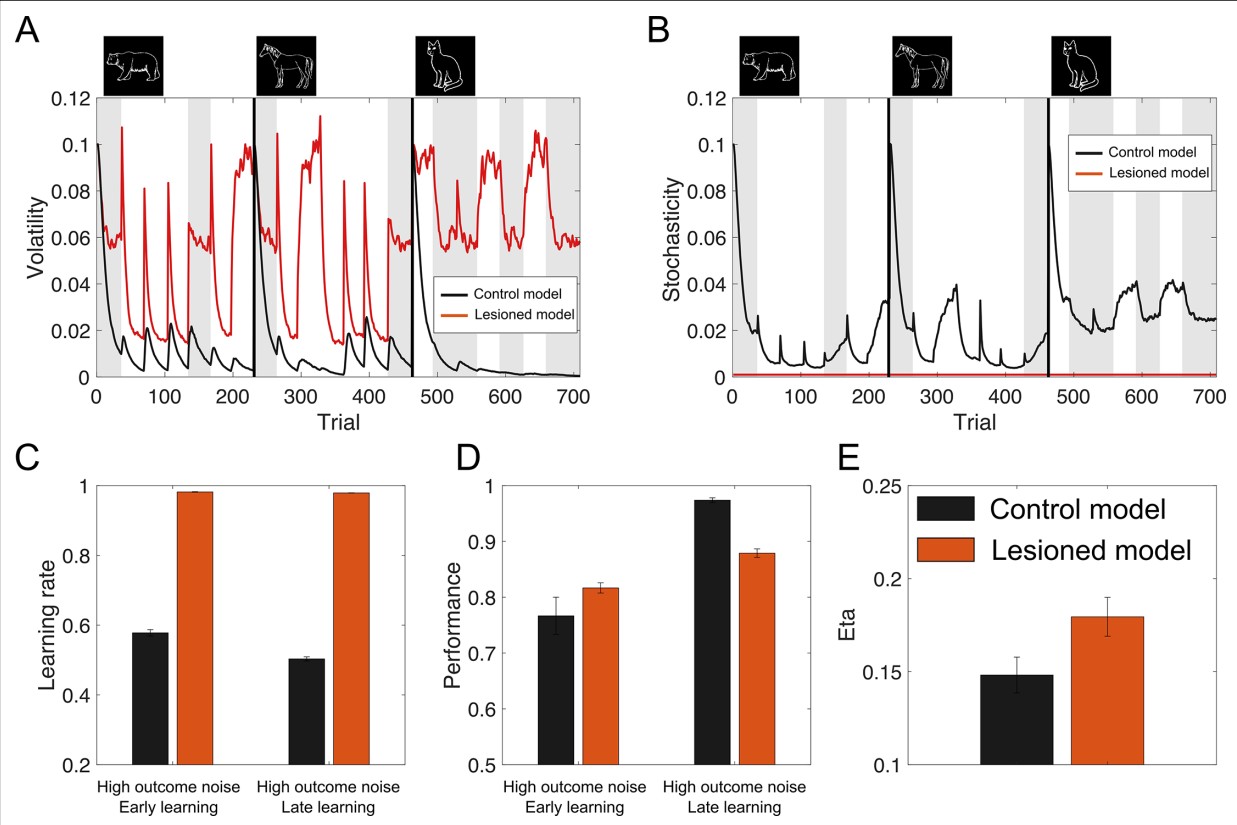

**Figure 9.** The stochasticity lesion model shows a pattern of learning deficits associated low performer in our task. Behavior of the lesioned model, in which stochasticity is assumed to be small and constant, is shown along the control model that jointly estimates stochasticity and volatility. (**A, B**) The inability to make inference about stochasticity, leads to misestimation of volatility, particularly for high outcome noise phases (gray patches show trials with high outcome noise 30/70% reward probability). (**C**) This led to reduced sensitivity of the learning rate to volatility (i.e., the first 10 trials after reversals). (**D**) The lesioned model shows similar behavior to our low-performer group, with reduced accuracy in later learnings stages for stimuli with high outcome noise. (**E**) When we fit simulated data (n = 100) from the two models to our model, we see increased eta parameter estimates for the lesioned model. Errorbars reflect standard error of the mean over 100 simulations.

The online version of this article includes the following figure supplement(s) for figure 9:

**Figure supplement 1.** Full model results.

**Figure supplement 2.** Simulated data from the stochasticity lesion model shows increased eta parameters compared to the control model.

However, misleading feedback (i.e., outcome noise) could be misinterpreted as indicating another change point because participants do not know the volatility beforehand. Strongly overinterpreting outcome noise as change points will hinder building a correct estimate of volatility and understanding the true structure of the task. Simultaneously estimating poses a challenge, as both contribute to greater outcome variance, making outcomes more surprising. A critical distinction, however, lies in their impact on generated outcomes: volatility increases the autocorrelation between consecutive outcomes, whereas stochasticity reduces it. Recent computational approaches have successfully utilized this fundamental difference to formulate a model of learning based on the joint estimation of stochasticity and volatility (*Piray and Daw, 2021*; *Piray and Daw, 2024*). They report evidence that humans successfully dissociate between volatility and stochasticity with contrasting and adaptive effects on learning rates, albeit to varying degrees. Interestingly, they show that hypersensitivity to outcome noise, often observed in anxiety disorders, might arise from a misattribution of the outcome noise to volatility instead of stochasticity resulting in increased learning rates and overadjustments to misleading outcomes. It is noteworthy, that we observed a similar hypersensitivity to high outcome noise in low performer in our task that is partly reduced by MA. In an exploratory analysis, we fit two models to our task structure using modified code provided by *Piray and Daw, 2021* (see Methods for formal Description of the model). The control model made inference about both the volatility and

stochasticity. The lesioned model assumed stochasticity to be small and constant. We show the results of this analyses in *Figure 9*, *Figure 9—figure supplements 1 and 2*. We found that the inability to make inference about stochasticity, leads to misestimation of volatility, particularly for high outcome noise phases (*Figure 9A, B*). Consistently, this led to reduced sensitivity of the learning rate to volatility (i.e., the first 10 trials after reversals). The model shows similar behavior to our low-performer group, with reduced accuracy in later learnings stages for stimuli with high outcome noise (*Figure 9D*). Finally, when we fit simulated data from the two models to our model, we see increased eta parameter estimates for the lesioned model. Together, these results may hint toward an overinterpretation of stochasticity in low performer of our task and that MA has beneficial effects for those individuals as it reduced the oversensitivity to volatility. It should be noted however, that we did not fit these models to our choice behavior directly as this implementation is beyond the scope of our current study. Yet, our exploratory analyses make testable predictions for future research into the effect of catecholamines on the inference of volatility and stochasticity.

### Control analyses

To control for the potentially confounding factor session order (i.e., PL first vs. MA first), we repeated the two-way mixed ANOVAs with significant Drug x Baseline Session interactions with session order as a between subject factors. Including session order did not alter the significance of the observed effects and did not interact with the effects of interest (all p > 0.24).

## Discussion

To study learning dynamics participants completed a reversal variant of an established probabilistic learning task (*Fischer and Ullsperger, 2013*; *Jocham et al., 2014*; *Kirschner et al., 2022*; *Kirschner et al., 2024*). Participants completed the task three times: in a baseline session without drug, and

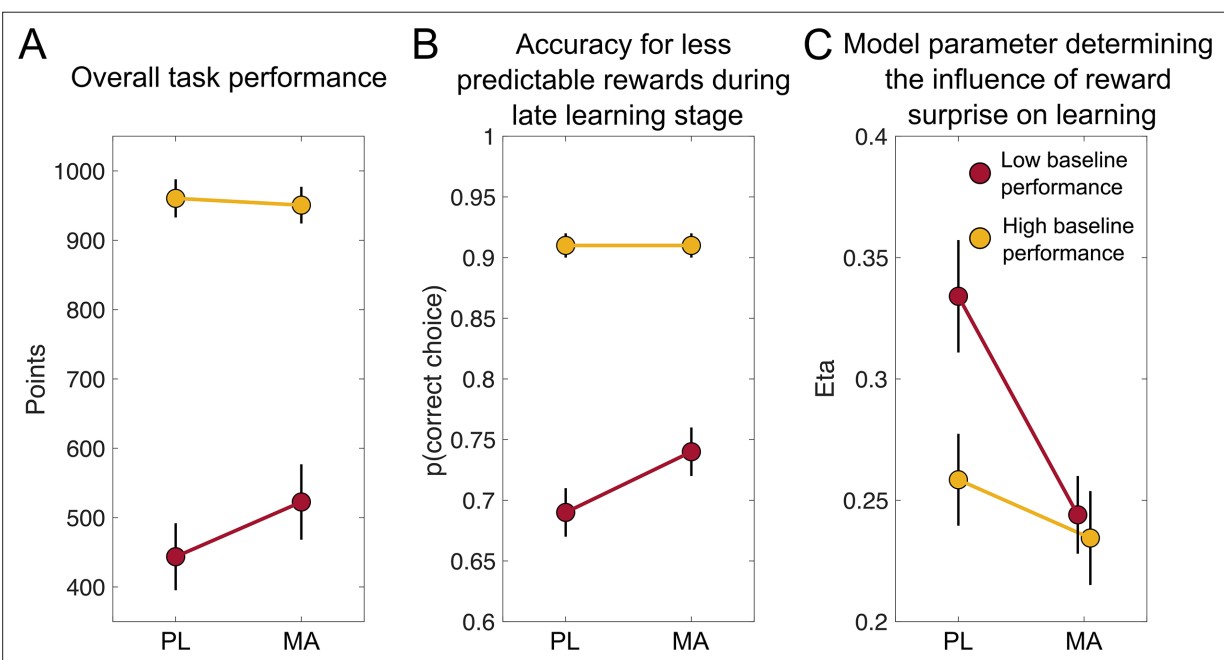

**Figure 10.** Summary of key findings. Mean (SEM) scores on three measures of task performance after PL and methamphetamine (MA), in participants stratified on low (n = 47) or high (n = 47) baseline performance. (**A**) There was a trend toward a drug effect, with boosted task performance (total points scored in the task) in low baseline performers (subjects were stratified via median split on baseline performance) after MA (20 mg) administration. (**B**) Follow-up analyses revealed that on-drug performance benefits were mainly driven by significantly better choices (i.e., choosing the advantageous stimuli and avoiding disadvantageous stimuli) at later stages after reversals for less predictable reward contingencies (30/70% reward probability). (**C**) To understand the computational mechanism through which MA improved performance in low baseline performers we investigated how performance in the task related to model parameters from our fits. Our results suggest that MA alters performance by changing the degree to which learning rates are adjusted according to recent prediction errors (eta), in particular by reducing the strength of such adjustments in low baseline performers to push them closer to task-specific optimal values.

after PL and after oral MA (20 mg) administration. We observed a trend toward a drug effect on overall performance, with improved task performance (total points scored in the task) selectively in low baseline performers. Follow-up analyses revealed that MA performance benefits were mainly driven by significantly better choices (i.e., choosing the advantageous stimuli and avoiding disadvantageous stimuli) at later stages after reversals for less predictable reward contingencies. Modeling results suggest that MA is helping performance by adaptively shifting the relative weighting of surprising outcomes based on their statistical context. Specifically, MA facilitated down-weighting of probabilistic errors in phases of less predictable reward contingencies. In other words, in low baseline performers the SNR between true reversals and misleading feedback is improved after the administration of MA. Moreover, although existing literature has linked catecholamines to volatility-based learning rate adjustments (*Cook et al., 2019*), we show that these adjustments also relate to other context-dependent adjustments like levels of probabilistic noise. The key findings of this study are summarized in *Figure 10*.

## MA affects the relative weighting of reward prediction errors

A key finding of the current study is that MA affected the relative weighting of reward prediction errors. In our model, adjustments in learning rate are afforded by weighting the learning rate as a function of the absolute value of the previous prediction error (*Li et al., 2011*). This associability-gated learning mechanism is empirically well supported (*Le Pelley, 2004*) and facilitates decreasing learning rates in periods of stability and increasing learning rates in periods of change. MA was associated with lower weighting of prediction errors (quantified by lower eta parameters under MA). Our results comprise an important next step in understanding the neurochemical underpinnings of learning rate adjustments.

Neuro-computational models suggest that catecholamines play a critical role in adjusting the degree to which we use new information. One class of models highlights the role of striatal dopaminergic prediction errors as a teaching signal in cortico-striatal circuits to learn task structure and rules (*Badre and Frank, 2012*; *Collins and Frank, 2013*; *Collins and Frank, 2016*; *Lieder et al., 2018*; *Pasupathy and Miller, 2005*; *Schultz et al., 1997*). The implication of such models is that learning the structure of a task results in appropriate adjustments in learning rates. Optimal learning in our task with high level of noise in reward probabilities in combination with changing reward contingencies required increased learning from surprising events during periods of change (reversals) and reduced learning from probabilistic errors. Thus, neither too low learning adjustments after surprising outcomes (low eta) nor too high learning adjustments after surprising outcomes (high eta) are beneficial in our task structure. Interestingly, MA appears to shift eta closer to the optimum. Exploratory simulation studies using a model that jointly estimates stochasticity and volatility (*Piray and Daw, 2021*; *Piray and Daw, 2024*), revealed that MA might reduce the oversensitivity to volatility. In terms of the neurobiological implementation of this effect, MA may prolong the impact of phasic dopamine signals, which in turn facilitates better learning of the task structure and learning rate adjustments (*Cook et al., 2019*; *Marshall et al., 2016*; *Volkow et al., 2002*). Our data, in broad strokes, are consistent with the idea that dopamine in the prefrontal cortex and basal ganglia is involved in modulating meta-control parameters that facilitated dynamic switching between complementary control modes (i.e., shielding goals from distracting information vs. shifting goals in response to significant changes in the environment) (*Cools, 2008*; *Dreisbach et al., 2005*; *Floresco, 2013*; *Goschke, 2013*; *Goschke and Bolte, 2014*; *Goschke and Bolte, 2018*). A key challenge in our task is differentiating real reward reversals from probabilistic misleading feedback which is a clear shielding/shifting dilemma described in the meta-control literature. Our data suggest that MA might improve meta-control of when to shield and when to shift beliefs in low baseline performers.

Moreover, it is possible that MA's effect on learning rate adjustments is driven by its influence on the noradrenaline system. Indeed, a line of research is highlighting the importance of the locus coeruleus/norepinephrine system in facilitating adaptive learning and structure learning (*Razmi and Nassar, 2022*; *Silvetti et al., 2018*; *Yu et al., 2021*). In particular, evidence from experimental studies, together with pharmacological manipulations and lesion studies of the noradrenergic system suggest that noradrenaline is important for change detection (*Muller et al., 2019*; *Nassar et al., 2012*; *Preuschoff et al., 2011*; *Set et al., 2014*). Thus, the administration of MA may have increased participants'

synaptic noradrenaline levels and, therefore, increased the sensitivity to salient events indicating true change points in the task.

It should be noted that other neuromodulators, such as acetylcholine (*Marshall et al., 2016*; *Yu and Dayan, 2005*) and serotonin (*Grossman et al., 2022*; *Iigaya et al., 2018*), have also been associated with dynamic learning rate adjustment. Future studies should compare the effects of neuromodulator-specific drugs for example a dopaminergic modulator, a noradrenergic modulator, a cholinergic modulator, and a serotonin modulator to make neuromodulator-specific claims (e.g., see *Marshall et al., 2016*). Taken together, it is likely that in our study MA effects on learning rate adjustments are driven by multiple processes that perhaps also work in concert. Moreover, because we only administered a single pharmacological agent, our results could reflect general effects of neuromodulation.

Our results are in line with recent studies that show improved performance under methylphenidate (MPH) by making learning more robust against misleading information. For example, *Fallon et al., 2017* showed that MPH helped participants to ignore irrelevant information but impaired the ability to flexibly update items held in working memory. Another study showed that MPH improved performance by adaptively reducing the effective learning rate in participants with higher working memory capacity (*Rostami Kandroodi et al., 2021*). These studies highlight the complex effects of MPH on working memory and the role of working memory in reinforcement learning (*Collins and Frank, 2012*; *Collins and Frank, 2018*). It could be that the effect of MA on learning rate dynamics reflect a modulation of interactions between working memory and reinforcement learning strategies. However, it should be acknowledged that our task was not designed to parse out specific contributions of the reinforcement learning system and working memory to performance.

## MA performance enhancement depends on initial task performance

Another key finding of the current study is that the benefits of MA on performance depend on the baseline task performance. Specifically, we found that MA selectively improved performance in participants that performed poorly in the baseline session. However, it should be noted, that all drug x baseline performance interactions, including for the key computational eta parameter did not reach the statistical threshold, and only tended toward significance. We used a binary discretization of baseline performance to simplify the analysis and presentation. To parse out the relationship between MA effects and baseline performance into finer level of detail, we conducted additional linear mixed-effects model (LMM) analyses using a sliding window regression approach (see Appendix 2). A key thing to notice in the sliding regression results is that, while each regression reveals that drug effects depend on baseline performance, they do so nonlinearly, with most variables of interest showing a saturating effect at low baseline performance levels and the strongest slope (dependence on baseline) at or near the median level of baseline performance, explaining why our median splits were able to successfully pick up on these baseline-dependent effects. Together, these results suggest that MA primarily affects moderately low baseline performer. It is noteworthy to highlight again that we had a separate baseline measurement from the PL session, allowing us to investigate baseline-dependent changes while avoiding typical concerns in such analyses like regression to the mean (*Barnett et al., 2005*). This design enhances the robustness of our baseline-dependent effects. It is important to note, that MA did not bring performance of low baseline performers to the level of performance of high baseline performers. We speculate that high performers gained a good representation of the task structure during the orientation practice session, taking specific features of the task into account (change point probabilities, noise in the reward probabilities). This is reflected in a large SNR between real reversals and misleading feedback. Because the high performers already perform the task at a near-optimal level, MA may not further enhance performance (see Appendix 4 for additional evidence for this claim). Intriguingly, the data do not support an inverted-U-shaped effect of catecholaminergic action (*Durstewitz and Seamans, 2008*; *Goschke and Bolte, 2018*) given that performance of high performers did not decrease with MA. One could speculate that catecholamines are not the only factor determining eta and performance. Perhaps high performers have a generally more robust/resilient decision-making system which cannot be perturbed easily. Probably one would need even higher doses of MA (with higher side effects) to impair their performance.

These results have several interesting implications. First, a novel aspect of our design is that, in contrast to most pharmacological studies, participants completed the task during a baseline session before they took part in the two drug sessions. Drug order and practice effects are typical nuisance

regressors in pharmacological imaging research. Yet, although practice effects are well acknowledged in the broader neuromodulator and cognitive literature (*Bartels et al., 2010*; *MacRae et al., 1988*; *Servan-Schreiber et al., 1998*), our understanding of these effects is limited. One of the few studies that report on drug administration effects, showed that d-amphetamine (AMPH) driven increases in functional-MRI-based blood oxygen level-dependent (BOLD) signal variability ($SD_{BOLD}$) and performance depended greatly on drug administration order (*Garrett et al., 2015*). In this study, only older subjects who received AMPH first improved in performance and $SD_{BOLD}$. Based on research in rats, demonstrating that dopamine release increases linearly with reward-based lever press practice (*Owesson-White et al., 2008*), the authors speculate that practice may have shifted participants along an inverted-U-shaped dopamine performance curve (*Cools and D'Esposito, 2011*) by increasing baseline dopamine release (*Garrett et al., 2015*). Interestingly, we did not see a modulation of the MA effects by drug session order (PL first vs. MA first). Thus, the inclusion of an orientation session might be a good strategy to control for practice and drug order effects.

Our results also illustrate the large interindividual variability of MA effects. Recently a large pharmacological fMRI/PET study ($n$ = 100) presented strong evidence that interindividual differences in striatal dopamine synthesis capacity explain variability in effects of methylphenidate on reversal learning (*van den Bosch et al., 2022*). They demonstrated that methylphenidate improved reversal learning performance to a greater degree in participants with higher dopamine synthesis capacity, thus establishing the baseline dependency principle for methylphenidate. These results are in line with previous research showing that methylphenidate improved reversal learning to a greater degree in participants with higher baseline working memory capacity, an index that is commonly used as an indirect proxy of dopamine synthesis capacity (*Rostami Kandroodi et al., 2021*; *van der Schaaf et al., 2013*; *van der Schaaf et al., 2014*). In the current study, we did not collect working memory capacity related information. However, our result that initial task performance strongly affected the effect of MA is in line with the pattern of results showing that individual baseline differences strongly influence drug effects and thus should be considered in pharmacological studies (*Cools and D'Esposito, 2011*; *Durstewitz and Seamans, 2008*; *van den Bosch et al., 2022*). Indeed, there is evidence from the broader literature on the effects of psychostimulants on cognitive performance, that suggest that stimulants improve performance only in low performers (*Ilieva et al., 2013*). Consistent with this, there is evidence in rats, that poor baseline performance was associated with greater response to amphetamine and increased performance in signal detection task (*Turner et al., 2017*).

Our findings can be contrasted to those of *Rostami Kandroodi et al., 2021*, who examined the effects of methylphenidate on a reversal learning task, in relation to baseline differences on a cognitive task. Whereas *Rostami Kandroodi et al., 2021* found that methylphenidate improved performance mainly in participants with higher baseline working memory performance, we found that MA improved the ability to dynamically adjust learning from prediction errors to a greater extent in participants who performed poorly to medium at baseline. There are several possible reasons for these apparently different findings. First, MA and methylphenidate differ in their primary mechanisms of action: MPH acts mainly as a reuptake blocker, whereas MA increases synaptic levels of catecholamines by inhibiting the vesicular monoamine transporter 2 and inhibiting the enzyme monoamine oxidase. These differences in action could account for differential effects on cognitive tasks. Second, the tasks used by *Rostami Kandroodi et al., 2021* and the present study differ in several ways. The *Rostami Kandroodi et al., 2021* task assessed responses to a single reversal event during the session, whereas the present study used repeated reversals with probabilistic outcomes. Third, the measures of baseline function differed in the two studies: *Rostami Kandroodi et al., 2021* used a working memory task that was not used in the drug sessions, whereas we used the probabilistic learning task as both the baseline measure and the measure of drug effects. Further research is needed to determine which of these factors influenced the outcomes.

## Conclusion

The current data provide evidence that relative to PL, MA facilitates the ability to dynamically adjust learning from prediction errors. This observation was seen to a greater degree in those participants who performed moderately low at baseline. These results advance existing literature by presenting evidence for a causal link between catecholaminergic modulation and learning flexibility and further highlight a baseline dependency principle for catecholaminergic modulation.

## Materials and methods

### Design

The results presented here were obtained from the first two sessions of a larger four-session study ( clinicaltrials.gov ID number NCT04642820). During the latter two sessions of the larger study, not reported here, participants participated in two fMRI scans. During the two 4 hr laboratory sessions presented here, healthy adults received MA (20 mg oral) or PL, in mixed order under double-blind conditions. One hour after ingesting the capsule, they completed the 30 min reinforcement reversal learning task. The primary comparisons were on acquisition and reversal learning parameters of reinforcement learning after MA vs. PL. Secondary measures included subjective and cardiovascular responses to the drug.

### Orientation session

Participants attended an initial orientation session to provide informed consent and to complete personality questionnaires. They were told that the purpose of the study was to investigate the effects of psychoactive drugs on mood, brain, and behavior. To reduce expectancies, they were told that they might receive a PL, stimulant, or sedative/tranquilizer. However, participants only received MA and PL. They agreed not to use any drugs except for their normal amounts of caffeine for 24 hr before and 6 hr following each session. Women who were not on oral contraceptives were tested only during the follicular phase (1–12 days from menstruation) because responses to stimulant drugs are dampened during the luteal phase of the cycle (*White et al., 2002*). Most participants (*N* = 97 out of 113) completed the reinforcement learning task during the orientation session as a baseline measurement. This measure was added after the study began. Participants who did not complete the baseline measurement were omitted from the analyses presented in the main text. We run the key analyses on the full sample (*n* = 109). This sample included participants who completed the task only on the drug sessions. When controlling for session order and number (two vs. three sessions) effects, we see no drug effect on overall performance and learning. Yet, we found that eta was also reduced under MA in the full sample, which also resulted in reduced variability in the learning rate (see Appendix 5 for more details).

### Drug sessions

The two drug sessions were conducted in a comfortable laboratory environment, from 9 am to 1 pm, at least 72 hr apart. Upon arrival, participants provided breath and urine samples to test for recent alcohol or drug use and pregnancy (CLIAwaived Inc, Carlsbad, CA Alcosensor III, Intoximeters; AimStickPBD, hCG professional, Craig Medical Distribution). Positive tests lead to rescheduling or dismissal from the study. After drug testing, subjects completed baseline mood measures, and heart rate and blood pressure were measured. At 9:30 am, they ingested capsules (PL or MA 20 mg, in color-coded capsules) under double-blind conditions. Oral MA (Desoxyn, 5 mg per tablet) was placed in opaque size 00 capsules with dextrose filler. PL capsules contained only dextrose. Subjects completed the reinforcement learning task 60 min after capsule ingestion. Drug Effects Questionnaires (DEQs) were obtained at multiple intervals during the session. They completed four other cognitive tasks not reported here. Participants were tested individually and were permitted to relax, read, or watch neutral movies when they were not completing study measures.

### Dependent measures

#### Reinforcement learning task

Participants performed a reversal variant of an established probabilistic learning task (*Fischer and Ullsperger, 2013*; *Jocham et al., 2014*; *Kirschner et al., 2022*; *Kirschner et al., 2024*). On each trial, participants were presented with one of three different stimuli and decided to either gamble or avoid gambling with that stimulus with the goal to maximize the final reward (see *Figure 2A*). A gamble resulted in winning or losing points, depending on reward contingencies associated with the particular stimulus. If participants decided not to gamble, they avoided any consequences but were still able to observe what would have happened if they had gambled by receiving counterfactual feedback. The three stimuli—white line drawings of animals on a black background—were presented in a pseudo-random series that was the same for all participants. Reward contingencies for every stimulus could be

20%, 30%, 50%, 70%, or 80% and stayed constant within one block of 30–35 trials. After every block, the reward contingency changed without notice. The experiment consisted of 7 blocks per stimulus, leading to 18 reversals and 714 trials in total (see Appendix 1 for additional information on time-on-task effects). Presentation 22.0 (Neurobehavioral Systems) was used for task presentation. Every trial of the task began with a central fixation cross, presented for a variable time between 300 and 500 ms. After fixation, the stimulus was presented together with the two choice alternatives (a green check-mark for choosing and a red no-go sign for avoiding, sides counterbalanced across subjects) for a maximum of 2000 ms or until a response was given. If participants failed to respond in time, a question mark was shown, and the trial was repeated at the end of the block. When a response was made, the stimulus stayed on screen, and feedback was given after 500 ms. The outcome was then presented for 750 ms depending on the subject's choice. Choosing to gamble led to either a green smiley face and a reward of 10 points or a red frowning face and a loss of 10 points according to the reward proba-bility of the stimulus. An avoided gamble had no monetary consequences: the outcome was always 0. Counterfactual/fictive outcomes, indicating what would have happened had the participant chosen to gamble, were shown on screen using the same smileys in a paler color, but the reward or punishment was crossed out to indicate that the outcome was fictive.

## Drug Effects Questionnaire (*Morean et al., 2013*)

The DEQ consists of five questions in total. In this paper, we only reported the ratings of the 'Do you feel any drug effect?' question which was rated on a 100-mm visual analog scale. Participants completed this at regular intervals throughout the session.

## Reinforcement learning model fitting

We fit variants of reinforcement learning models to participants' choice behavior using a constrained search algorithm (fmincon in MATLAB 2021b), which computed a set of parameters that maximized the total log posterior probability of choice behavior. The base model (M1) was a standard Q-learning model with three parameters: (1) an inverse temperature parameter of the softmax function used to convert trial expected values to action probabilities, (2) a play bias term that indicates a tendency to attribute higher value to gambling behavior (*Jang et al., 2019*), and (3) an intercept term for the effect of learning rate on choice behavior (*Jang et al., 2019*). On each trial, the expected value ($Q_t$) of a stimulus ($X_t$) was calculated according to the following formula:

$$Q_{t+1}\left(X_t\right) = Q_t\left(X_t\right) + \alpha * \delta_t \ with \ \delta_t = R_t - Q_t\left(X_t\right)$$

$Q$ values represent the expected value of an action at trial $t$. $\alpha$ reflects the learning rate. $\delta_t$ represents the prediction error with $R_t$ being the reward magnitude of that trial. On each trial, this value term was transferred into a 'biased' value term ($V_B\left(X_t\right) = B_{play} + Q_t\left(X_t\right)$, where $B_{play}$ is the play bias term) and converted into action probabilities $P(play|V_{B\,play}(t)(X_t))$; $P(pass|V_{B\,pass}(t)(X_t))$ using a softmax function with an inverse temperature ($\beta$):

$$P\left(\text{play}|\ V_{Bplay}\left(t\right)\left(X_t\right)\right) = \frac{\exp\left(V_{\text{B play}}\left(t\right)\left(X_t\right)\cdot\beta\right)}{\exp\left(V_{\text{play}}\left(t\right)\left(X_t\right)\cdot\beta\right) + \exp\left(V_{\text{pass}}\left(t\right)\left(X_t\right)\cdot\beta\right)},$$

$$P\left(\text{pass}|V_{Bpass}\left(t\right)\left(X_t\right)\right) = \frac{exp\left(V_{\text{pass}}\left(t\right)\left(X_t\right)\cdot\beta\right)}{exp\left(V_{\text{play}}\left(t\right)\left(X_t\right)\cdot\beta\right) + exp\left(V_{\text{pass}}\left(t\right)\left(X_t\right)\cdot\beta\right)}.$$

This was our base model (M1). Next, we fit further reinforcement models by complementing the base model with additional parameters. These additional parameters controlled trial-by-trial modu-lations of the learning rate. Note that our base model treats the learning rate for value updates as a constant. However, previous studies have shown that people are able to adjust their learning rate according to the volatility of the environment (*Behrens et al., 2007*; *Nassar et al., 2010*). In the Pearce–Hall hybrid model, adjustments in learning rate are afforded by weighting the learning rate as a function of the absolute value of previous prediction error (*Li et al., 2011*). This associability-gated learning mechanism is empirically well supported (*Le Pelley, 2004*) and facilitates decreasing learning rates in periods of stability and increasing learning rates in periods of change. Previous work has

shown that the hybrid model can approximate normative learning rate adjustments (*Li et al., 2011*; *Piray et al., 2019*). In this hybrid model, the learning rate is updated as follows:

$$\alpha_t = \kappa A_t,$$

$$A_{t+1}\left(X_t\right) = \eta * \left|\delta_t\right| + \left(1 - \eta\right) * A_t\left(X_t\right).$$

Here, $\kappa$ is scale of learning rate ($\alpha_t$ and $\eta$) determines the step size for updating associability ($A_t$) as a function of the absolute RPE ($\left|\delta_t\right|$). On each trial, the learning rate $\alpha_t$ depends on the absolute RPE from the past trial. Note that the initial learning rate is defined by $\kappa$, whereby $\kappa$ is determined by a logistic function of a weighted predictor matrix that could include an intercept term (Pearce–Hall hybrid model (M2)) and task variables that may additionally affect trial-by-trial learning rate adjustments. In the Pearce–Hall hybrid feedback confirmatory model (M3), the predictor matrix included an intercept term and feedback confirmatory information (i.e., was the feedback on a given trial confirmatory (factual wins and counterfactual losses) or disconfirmatory (factual losses and counterfactual wins)). Finally, in the Pearce–Hall hybrid feedback confirmatory and modality model (M4), the predictor matrix included an intercept term, feedback confirmatory information, and feedback modality (factual vs. counterfactual feedback) information. Two additional learning rate terms—feedback confirmation and modality—were added to the model set, as these factors have been shown to influence learning in similar tasks (*Kirschner et al., 2024*; *Schüller et al., 2020*). The best-fitting model was determined by computing the BIC for each model (*Schwarz, 1978*). Moreover, we computed protected exceedance probabilities, which give the probability that one model was more likely than any other model of the model space (*Rigoux et al., 2014*). To compare participant behavior to model-predicted behavior, we simulated choice behavior using the best-fitting model (Pearce–Hall hybrid feedback confirmatory model; see *Figure 3A*). For each trial, we used the expected trial value ($Q_t\left(X_t\right)$) computed above, and the parameter estimates of the temperature variable as inputs to a softmax function to generate choices. Validation of model selection and parameter recovery is reported in *Figure 4—figure supplement 1*.

## Description of the joint estimation of stochasticity and volatility model

In an exploratory analysis, we fit two models to our task structure using modified code provided by *Piray and Daw, 2021*. The following description of the model is adapted therefrom.

In this model, the outcome on trial t, denoted as $o_t$, is determined by three latent variables: reward probability rate, stochasticity, and volatility. The reward rate on trial t, represented as $x_t$, follows a Markov process with the following dynamics:

$$x_t = x_{t-1} + \epsilon_t$$

where $\epsilon_t$ is Gaussian noise with zero mean and a variance governed by volatility. Consequently, the probability distribution of $x_t$ given $x_{t-1}$ and volatility $v_t$ is:

$$p\left(x_t|x_{t-1}, v_t\right) = N\left(x_t|x_{t-1}, v_t\right),$$

where $v_t$ represents the volatility. For analytical convenience, the inverse volatility is defined as:

$$z_t = v_t^{-1},$$

a formulation commonly used in previous studies due to its computational advantages (*Piray and Daw, 2020*). Outcome values are generated based on the reward rate and stochasticity, following a Gaussian distribution:

$$p\left(o_t|x_t, s_t\right) = N\left(o_t|x_t, s_t\right),$$

where $s_t$ represents stochasticity, and the inverse stochasticity is denoted as:

$$y_t = s_t^{-1}.$$

For volatility and stochasticity, a multiplicative noise model was applied to their inverse, an approach previously shown to facilitate analytical inference when considered in isolation (*Gamerman et al., 2013*; *West, 1987*). Specifically, the dynamics governing these variables follow:

$$z_t = \eta_v^{-1} z_{t-1} \epsilon_t,$$

where $0 < \eta_v < 1$ is a constant, and $\epsilon_t$ is a random variable within the unit range, following a Beta distribution:

$$p\left(\epsilon_t\right) = B\left(\epsilon_t | j, 0.5\eta_v\left(1-\eta_v\right)^{-1}, 0.5\right).$$

The conditional expectation of $z_t$ is given by $z_{t-1}$, since $E\left(\epsilon_t\right) = \eta_v$. A similar and independent dynamic is assumed for $y$, parameterized by the constant $\eta_s$, such that:

$$y_t = \eta_s^{-1} y_{t-1} \varepsilon_t,$$

where $\varepsilon_t$ follows a distribution similar to $\varepsilon_t$, parameterized by $\eta_s$.

In this implementation, the model was parameterized using $\lambda_v = 1 - \eta_v$ and $\lambda_s = 1 - \eta_s$, respectively. These parameters are interpreted as the update rate for volatility and stochasticity. In other words, larger values of $\lambda_v$ and $\lambda_s$ lead to faster updates of volatility and stochasticity. Intuitively, a smaller $\lambda_v$ increases the mean of $\varepsilon_t$, resulting in a larger update of $z_t$. Since volatility is the inverse of $z_t$, a smaller $\lambda_v$ consequently leads to a slower update of volatility. This relationship has been formally demonstrated in recent work (*Piray and Daw, 2020*). In addition to these two parameters, the generative process depends on the initial values of volatility and stochasticity, denoted as $v_0$ and $s_0$. For our simulations, we assumed $v_0 = 0.1$ and $s_0 = 0.1$. For inference, a Rao-Blackwellized Particle Filtering approach (*Murphy and Russell, 2001*) was employed, where inference about $v_t$ and $s_t$ was performed using a particle filter (*Doucet and Johansen, 2011*), and conditional on these, inference over $x_t$ was carried out using the Kalman filter.

The Kalman filter maintains beliefs about the reward probability at each trial in the form of a Gaussian distribution, characterized by a mean $m_t$ and a variance $w_t$, which represents uncertainty about the true value. On each trial, the update is driven by a prediction error signal $\delta_t$ and a learning rate $\alpha_t$. This results in the following update rules upon observing the outcome $o_t$:

$$\delta_t = o_t - m_t$$
$$\alpha_t = \frac{w_t + v_t}{w_t + v_t + s_t}$$
$$m_{t+1} = m_t + \alpha_t \delta_t,$$
$$w_{t+1} = \left(1 - \alpha_t\right)\left(w_t + v_t\right).$$

These equations describe how the belief about the reward rate is adjusted dynamically based on observed outcomes and the associated uncertainty.

The particle filter is a Monte Carlo sequential importance sampling method that maintains a set of particles (i.e., samples). The algorithm consists of three steps per trial. First, in the prediction step, each particle transitions to the next state based on the generative process. Second, the weight of each particle is updated based on the probability of the observed outcome:

$$b_t^l \propto N\left(o_t | m_{t-1}^l, w_{t-1}^l + v_t^l + s_t^l\right),$$

where $b_t^l$ is the weight of particle $l$ on trial $t$, and $m_{t-1}^l$ and $w_{t-1}^l$ are the estimated mean and variance from the Kalman filter on the previous trial. The terms $v_t^l$ and $s_t^l$ represent the sampled values of volatility and stochasticity (i.e., the inverse of $z_t^l$ and $y_t^l$). During this step, particles are resampled using the systematic resampling procedure if the ratio of effective to total particles falls below 0.5. In the third step, the Kalman filter updates the mean and variance. Specifically, for every particle, the equations of the Kalman filter are used to define $alpha_t^l$ and update $m_t^l$ and $w_t^l$. The learning rate and estimated reward probability on trial $t$ are then computed as the weighted average of all particles, with the weights given by $b_t^l$.

In the simulations, the reward probability followed those of our task with outcome variance set to 0.01 for low outcome noise (20/80% reward probability), 0.02 for high outcome noise (30/70% reward probability), and 0.05 for random reward probabilities. The stochasticity for the lesioned models in the simulations was assumed to be 0.001. For simulating choice, we used the softmax with a decision noise of 3.

## Data analysis

We analyzed drug effects on behavioral performance and model parameters using paired *t* tests. Given the effects of initial performance and practice in pharmacological imaging research (*Garrett et al., 2015*), we additionally stratified MA effects by task performance in the orientation using median split. These data were analyzed using a two-way repeated-measures ANOVA with the factors Drug (two levels) and Baseline Performance (two levels). Paired *t*-tests were used as post hoc tests. Moreover, we investigated reversal learning by calculating learning curves. Post hoc, we observed that drug effects on learning became only apparent in the second phase of learning. We therefore used the Bai–Perrin multiple breakpoint test (*Bai and Perron, 2003*) to identify the number and location of structural breaks in the learning curves. In broad strokes, the test detects whether breaks in a curve exist, and if so, how many there are, based on the regression slope in predefined segments (here, we set the segment length to five trials). In our case, the test could reveal between 0 and 5 breaks (number of trials/segment length − 1). We run this test using data from all subjects and all sessions. The test detected one break that cut the learning curves into two segments (see results). We then calculated an index of learning performance after reversals by averaging the number of correct choices over the second learning phase. The index was then subjected to a two-way repeated ANOVA with the factors Drug (two levels) and Baseline Performance (two levels).

## Acknowledgements

We thank all our participants who took part in this research for the generosity of their time and commitment. This research was supported by the National Institute on Drug Abuse DA02812. HdW was supported by the National Institutes of Health T32 GM07019. MU was supported by the Deutsche Forschungsgemeinschaft, Grant/Award Number: SFB 1436; and the European Research Council, Grant/Award Number: 101018805.

## Additional information

### Competing interests

Harriet de Wit: is on the Board of Directors of PharmAla Biotech, and on scientific advisory committees of Gilgamesh Pharmaceuticals and MIND Foundation. These activities are unrelated to the present study. The other authors declare that no competing interests exist.

### Funding

| Funder | Grant reference number | Author |
| --- | --- | --- |
| National Institute on Drug Abuse | DA02812 | Harriet de Wit |
| National Institutes of Health | T32 GM07019 | Harriet de Wit |
| European Research Council | 10.3030/101018805 | Markus Ullsperger |
| Deutsche Forschungsgemeinschaft | SFB 1436 | Markus Ullsperger |

The funders had no role in study design, data collection, and interpretation, or the decision to submit the work for publication.

### Author contributions

Hans Kirschner, Conceptualization, Formal analysis, Visualization, Writing – original draft, Writing – review and editing; Hanna M Molla, Conceptualization, Visualization, Writing – original draft, Project administration, Writing – review and editing; Matthew R Nassar, Supervision, Methodology, Writing – original draft, Writing – review and editing; Harriet de Wit, Conceptualization, Resources, Supervision, Funding acquisition, Methodology, Writing – original draft, Project administration, Writing – review

and editing; Markus Ullsperger, Conceptualization, Resources, Supervision, Methodology, Writing – original draft, Writing – review and editing

## Author ORCIDs
Hans Kirschner https://orcid.org/0000-0001-9747-1746
Hanna M Molla https://orcid.org/0000-0002-2971-4512
Matthew R Nassar https://orcid.org/0000-0002-5397-535X
Harriet de Wit https://orcid.org/0000-0002-7211-8994
Markus Ullsperger https://orcid.org/0000-0003-3970-1982

## Ethics
Clinical trial registration clinicaltrials.gov ID number NCT04642820.
Informed consent, including consent to publish, was obtained from all participants. The study was approved by the Institutional Review Board of the University of Chicago.

Reviewer #1 (Public review): https://doi.org/10.7554/eLife.101413.3.sa1
Reviewer #2 (Public review): https://doi.org/10.7554/eLife.101413.3.sa2
Author response https://doi.org/10.7554/eLife.101413.3.sa3

## Additional files

### Supplementary files
MDAR checklist

### Data availability
All raw data and analysis scripts can be accessed at: https://github.com/HansKirschner/REFIT_Chicago_public (copy archived at *Kirschner, 2025*).

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

# Appendix 1

## Time-on-task effects

Given the length of our task, we investigated whether fatigue effects or changes in behavior occurred over time. Specifically, we regressed each participant's single-trial log-scaled reaction times (RT) and accuracy (a binary variable reflecting whether participants displayed stimulus-appropriate behavior on each trial) onto trial number, which served as a proxy for time on task. The resulting *t*-values for the time-on-task regressor were analyzed at the group level using two-sided *t*-tests against zero and compared across sessions and baseline performance groups. The results of these regression models are presented in *Appendix 1—table 1*, with raw data splits shown in *Appendix 1—figure 1*.

Our findings indicate that choice behavior was not systematically affected over the course of the task. This effect did not differ between low and high baseline performers and was not influenced by the drug. In contrast, reaction times decreased over the course of the task, with this speeding effect being enhanced by MA, particularly in the low performance group.

**Appendix 1—table 1.** Time-on-task effect on choice and reaction time.

| Model | ToT Coefficient PLMean (SD) | ToT Coefficient MAMean (SD) | Group comparison |
|---|---|---|---|
| Logistic regression on accuracy | | | |
| All | -0.05(0.17) | -0.31(0.23) | t(93)=0.90, *P*=0.37 |
| Low performer | -0.18(0.35) | 0.41(0.25) | t(46)=1.61, *P*=0.11 |
| High performer | -0.43(0.32) | -0.51(0.22) | t(46)=0.16, *P*=0.86 |
| Regression on log(RT) | | | |
| All | -1.77(0.42)* | -3.09 (0.48)* | t(93)=2.28, *P*=0.02 |
| Low performer | -2.11(0.65)* | -4.32(0.65)* | t(46)=−2.59, *P*=0.01 |
| High performer | -1.43(0.54)* | -1.85(0.64)* | t(46)=−0.55, *P*=0.58 |

Note: ToT = time-on-task.

*average within participant *t*-values significantly differ from zero.

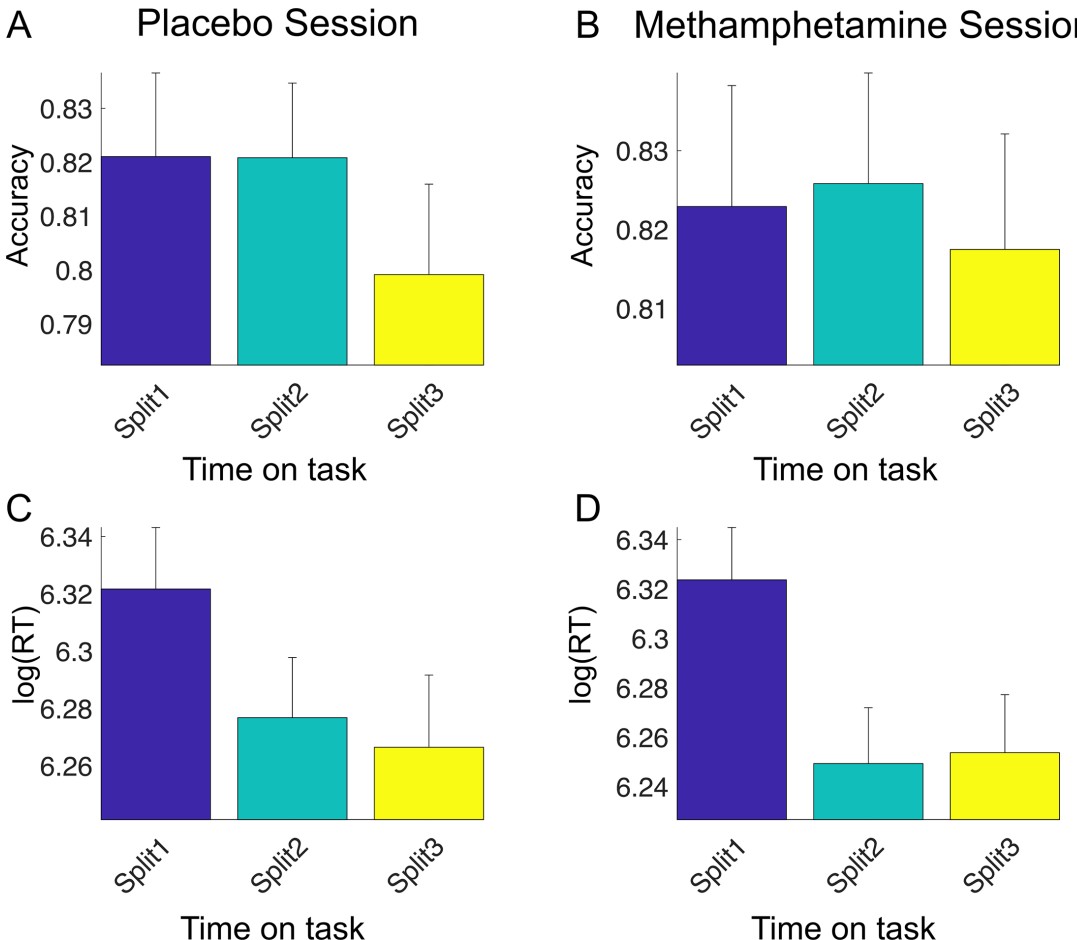

**Appendix 1—figure 1.** Speeding over the course of the task is enhanced by methamphetamine. This figure shows raw data splits for the effect of time on task (i.e., trial number) on accuracy (panels **A** and **B**; a binary variable indicating stimulus-appropriate behavior on each trial) and log-scaled reaction times (RT; panels **C** and **D**) across different drug sessions. Results indicate that overall choice accuracy was not affected by time on task. However, participants exhibited faster reaction times over the course of the task, with this speeding effect being more pronounced in the methamphetamine condition. Errorbars reflect standard error of the mean over 94 subjects.

## Appendix 2

## Linear mixed-effects model analyses for key findings

In the manuscript, we used a binary discretization of baseline performance to simplify the analysis and presentation. To parse out the relationship between methamphetamine effects and baseline performance into finer level of detail, we conducted additional linear mixed-effects model (LMM) analyses, focusing on the key findings reported in the manuscript. Specifically, we examined drug effects:

- On overall total points.
- On the probability of correct choice in the late learning phase under high outcome noise.
- On the learning rate parameter eta.
- On learning rate variability.
- On choices following double-misleading feedback.
- On the signal-to-noise ratio in high outcome noise conditions.

We first fit the following model:

$$\text{Outcome}_i = \beta_0 + \beta_1 \text{SessionOrder}_i + \beta_2 \text{DrugSession}_i + \beta_3 \text{BLPerformance}_i + \beta_4 \left(\text{DrugSession}_i \times \text{BLPerformance}_i\right) + u_{s(i)} + \varepsilon_i$$

where:

- $Outcome_i$ is the dependent variable (i.e., total points, $P$(correct choice) late learning phase high outcome noise; parameter estimate for eta; learning rate variance; $P$(correct choice) after double-misleading outcomes; signal-to-noise ratio high outcome noise).
- $\beta_0$ is the fixed intercept.
- $\beta_1$ and $\beta_2$ are the fixed-effect coefficients.
- Session Order$_i$ (drug first vs. placebo first) is a fixed effect to control for possible session-order effects.
- Drug Session$_i$ is a fixed effect for drug session.
- $u_{s(i)} \sim N\left(0, \sigma_u^2\right)$ is the random intercept for subject $s\left(i\right)$.
- $\varepsilon_{s(i)} \sim N\left(0, \sigma_u^2\right)$ is the residual error term.

To our surprise, none of these models revealed a significant drug × baseline performance interaction. To further investigate this, we plotted the difference scores (i.e., the outcome variable under the drug condition minus placebo condition) against baseline performance scores. To better visualize potential nonlinearities, we applied a sliding window approach (step size of one subject and a window size of 25 subjects based on recent recommendations *Jenkins and Quintana-Ascencio, 2020*). This revealed nonlinear relationships between baseline performance and outcome variables (see *Appendix 2—figure 1*). Specifically, we observed that drug effects were maximal in moderately low baseline performers.

To formally test this pattern, we applied the same sliding window regression approach, fitting a set of LMMs separately for each step size. These models followed the same structure as before but excluded the baseline performance term, allowing us to examine within-window effects:

$$OutcomeVariable_i = \beta_0 + \beta_1 \text{SessionOrder}_i + \beta_2 \text{DrugSession}_i + u_{s(i)} + \varepsilon_i,$$

where

- $OutcomeVariable_i$ is the same as described above.
- $\beta_0$ is the fixed intercept.
- $\beta_1$ and $\beta_2$ are the fixed-effect coefficients.
- Session Order$_i$ and Drug Session$_i$ remain as fixed effects.
- $u_{s(i)} \sim N\left(0, \sigma_u^2\right)$ is the random intercept for subject $s\left(i\right)$.
- $\varepsilon_{s(i)} \sim N\left(0, \sigma_u^2\right)$ is the residual error term.

The results of these models (see *Appendix 2—figure 2*) indicate that indeed the drug effect is strongest in moderately low baseline performers. A key thing to notice in the sliding regression results is that, while each regression reveals that drug effects depend on baseline performance, they do so nonlinearly, with most variables of interest showing a saturating effect at low baseline performance levels and the strongest slope (dependence on baseline) at or near the median level of

baseline performance, explaining why our median splits were able to successfully pick up on these baseline-dependent effects.

Together, these results suggest that methamphetamine primarily affects individuals with medium-to-low baseline performance.

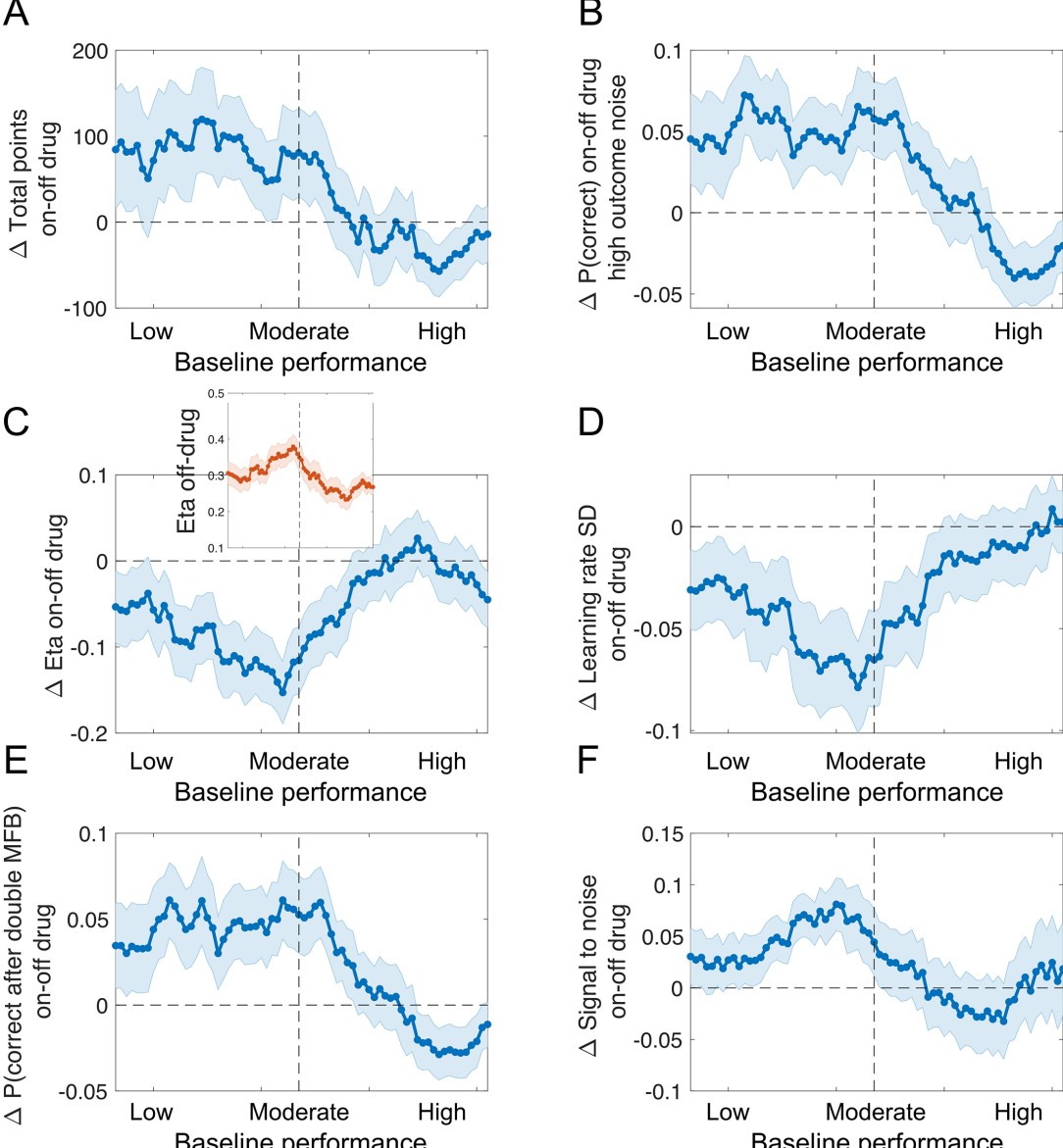

**Appendix 2—figure 1.** The effect of methamphetamine depends on baseline performance. To examine the relationship between methamphetamine effects and baseline performance, we plotted difference scores (i.e., the outcome variable under the drug condition minus placebo condition) against baseline performance. (**A–F**) show nonlinear relationships between baseline performance and outcome variables ((**A**) total points, (**B**) probability of a correct choice (*P*(correct choice)) in the late learning phase with high outcome noise, (**C**) parameter estimate for eta, (**D**) learning rate variance, (**E**) probability of a correct choice (*P*(correct choice)), and (**F**) signal-to-noise ratio in high outcome noise). Specifically, we show that drug effects were maximal in moderately low baseline performers. It is noteworthy that these subjects had particularly high eta's on placebo (inset in **C**), which may have allowed the drug effects to have a larger impact on their performance. This is in line with our key finding, that methamphetamine brings eta (parameter controlling dynamic adjustments of learning rate according to recent prediction errors) closer to optimal levels. Vertical dashed line indicates medium baseline performance. MFB = misleading feedback. Mean/SEM = line/shading.

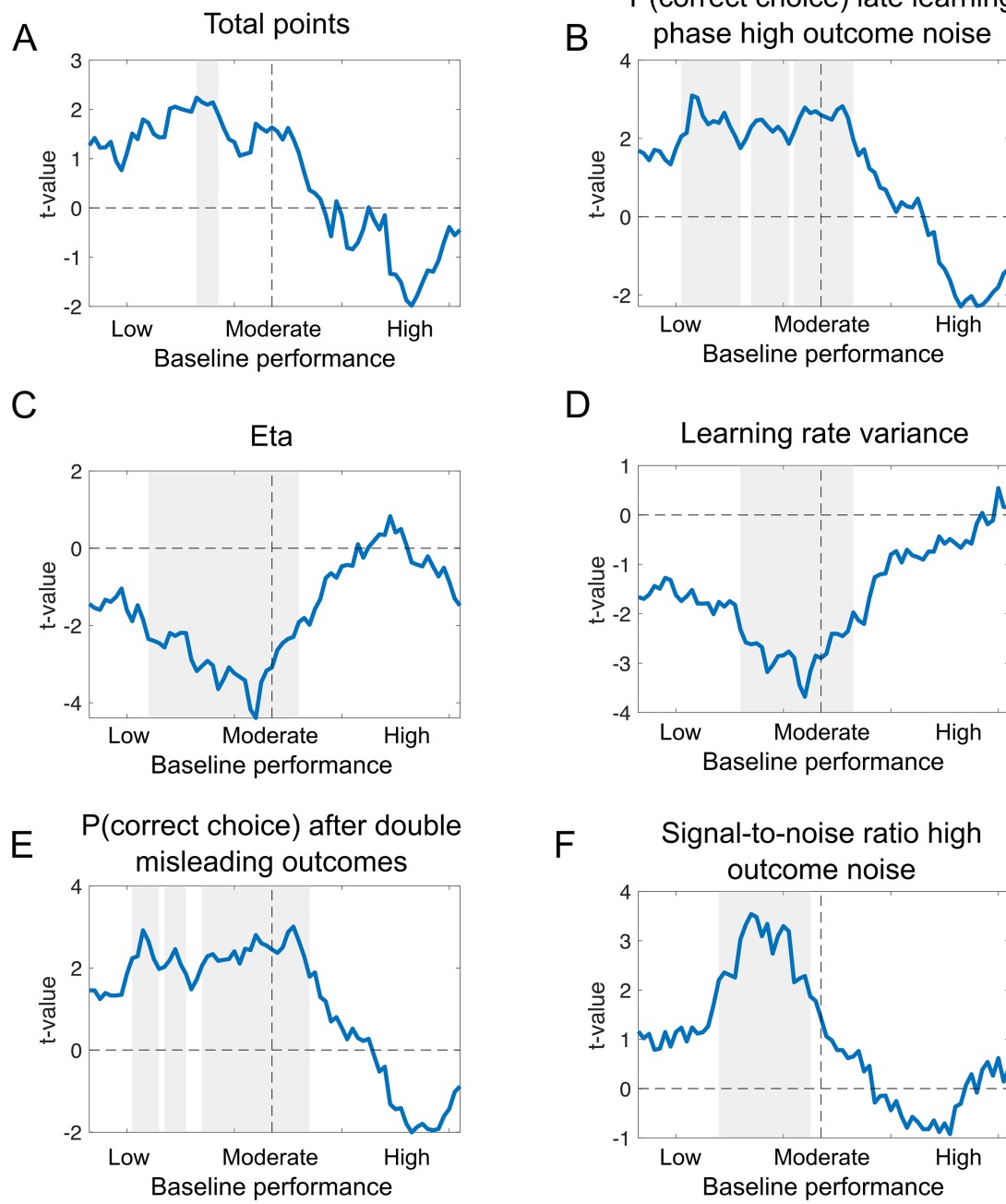

**Appendix 2—figure 2.** The effect of methamphetamine is strongest in moderately low baseline performers. This figure shows the drug effect from the sliding window linear mixed-effects model analysis plotted against baseline performance for a set of dependent variables: (**A**) total points, (**B**) probability of a correct choice (*P*(correct choice)) in the late learning phase with high outcome noise, (**C**) parameter estimate for eta, (**D**) learning rate variance, (**E**) probability of a correct choice (*P*(correct choice)), and (**F**) signal-to-noise ratio in high outcome noise. Gray areas indicate clusters with significant drug effects after correcting for multiple comparisons using a permutation test for cluster mass (*Maris and Oostenveld, 2007*).

# Appendix 3

## Analysis of parameter differences between methamphetamine (MA) and placebo (PL)

**Appendix 3—table 1.** Analysis of parameter differences between MA and PL.

| Model | Parameter PL Mean (SD) | Parameter MA Mean (SD) | Inferential statistic | Signed-Rank Test |
|---|---|---|---|---|
| Model $\Delta_{\text{inverse temperature}}$ | | | | |
| Low performer | 1.90 (0.16) | 1.83 (0.11) | $t(46) = 0.60, p=0.55$ | p = 0.899 |
| High performer | 3.12 (0.08) | 3.06 (0.08) | $t(46) = 1.00, p=0.32$ | p = 0.088 |
| Model $\Delta_{\text{play bias}}$ | | | | |
| Low performer | 0.04 (0.05) | 0.02 (0.03) | $t(46) = 0.26, p = 0.79$ | p = 0.168 |
| High performer | –0.03 (0.01) | –0.01 (0.02) | $t(46) = -1.03, p = 0.30$ | p = 0.189 |
| Model $\Delta_{\text{LR intercept}}$ | | | | |
| Low performer | –7.17 (0.58) | –7.24 (0.64) | $t(46) = 0.17, p = 0.86$ | p = 0.611 |
| High performer | –4.86 (0.15) | –4.92 (0.15) | $t(46) = 0.47, p = 0.63$ | p = 0.227 |
| Model $\Delta_{\text{eta}}$ | | | | |
| Low performer | 0.35 (0.02) | 0.28 (0.02) | $t(46) = 2.90, p = 0.005$ | p = 0.006 |
| High performer | 0.26 (0.02) | 0.24 (0.01) | $t(46) = 1.55, p = 0.12$ | p = 0.114 |

Here, we extended the winning model by allowing each parameter, in turn, to be differentially estimated for MA and PL, while keeping the other parameters fixed at the group (low and high baseline performance) mean estimates of the winning model for the placebo session.

## Appendix 4

## Near-optimal performance

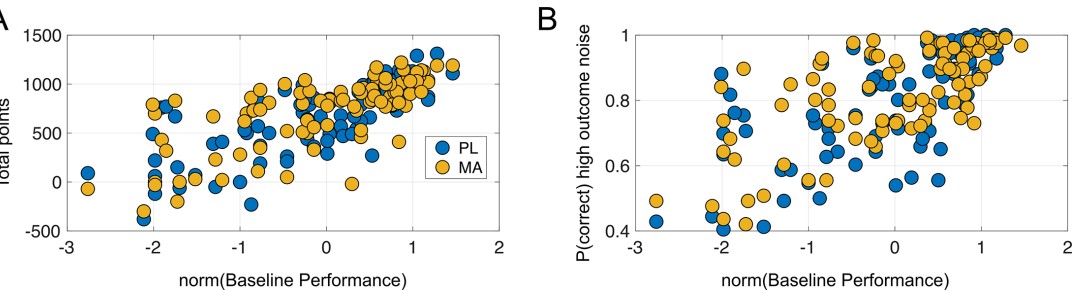

**Appendix 4—figure 1.** High baseline performers reach near-optimal levels in both test sessions, limiting potential drug-induced performance enhancement. (**A**) Overall task performance and (**B**) the probability of correct choices in the high outcome noise condition—the condition with the strongest observed drug effect—are plotted against normalized baseline performance. In both testing sessions, high baseline performers cluster around optimal performance. Furthermore, performance simulations using (**a**) optimal eta values and (**b**) observed eta values from the high baseline performance group reveal only a small, non-significant performance difference (points optimal eta: 701.91 (21.66) vs. points high performer: 694.47 (21.71); $t(46) = 2.84$, p = 0.07, $d = 0.059$; $t(46) = 2.84$, p = 0.07, $d = 0.059$). These results suggest that high baseline performers are already at or near their performance ceiling, limiting the potential for further drug-induced improvements.

## Appendix 5

### Full Sample results

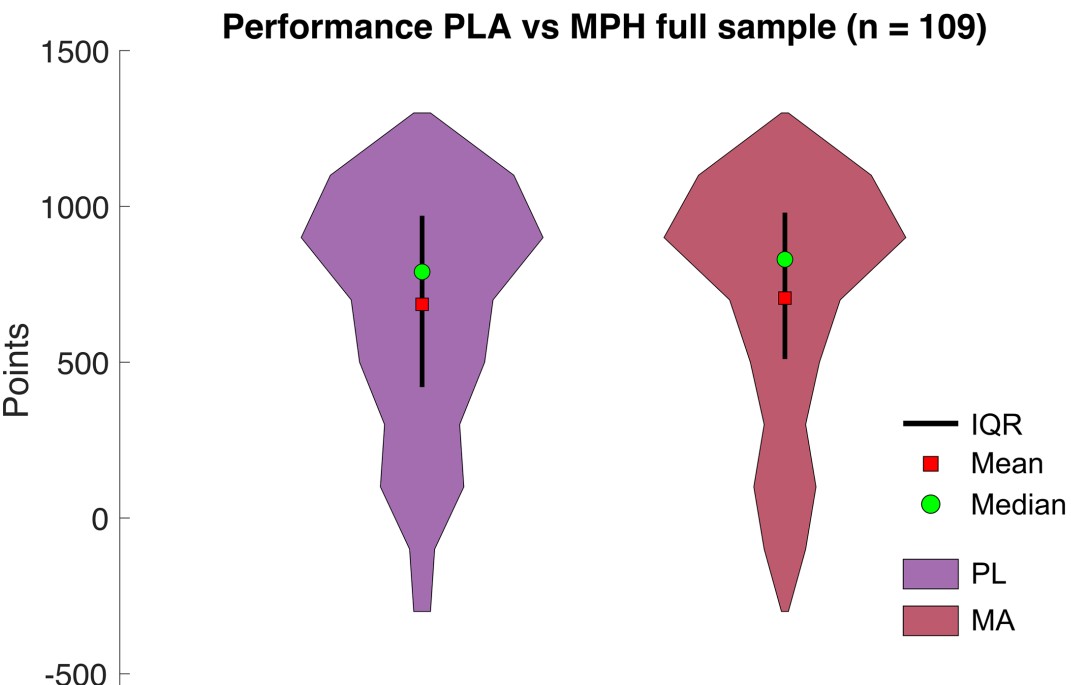

**Appendix 5—figure 1.** Overall points full sample. When comparing overall points in the whole sample (*n* = 109), we do not see a difference between MPH vs. PLA (705.68 (36.27) vs. 685.77 (35.78); *t*(108) = 0.81, p = 0.42, *d* = 0.05). Repeated mixed ANOVAs suggested that drug effects did not depend on session order (MPH first vs. PLA first), or whether subjects performed the orientation session. Yet, participants who completed the orientation tended to perform better during the drug sessions (*F*(1,107) = 3.09, p = 0.08; 719.31 (26.6264) vs. 548.00 (75.09)). *Note.* PL = placebo; MA = methamphetamine.

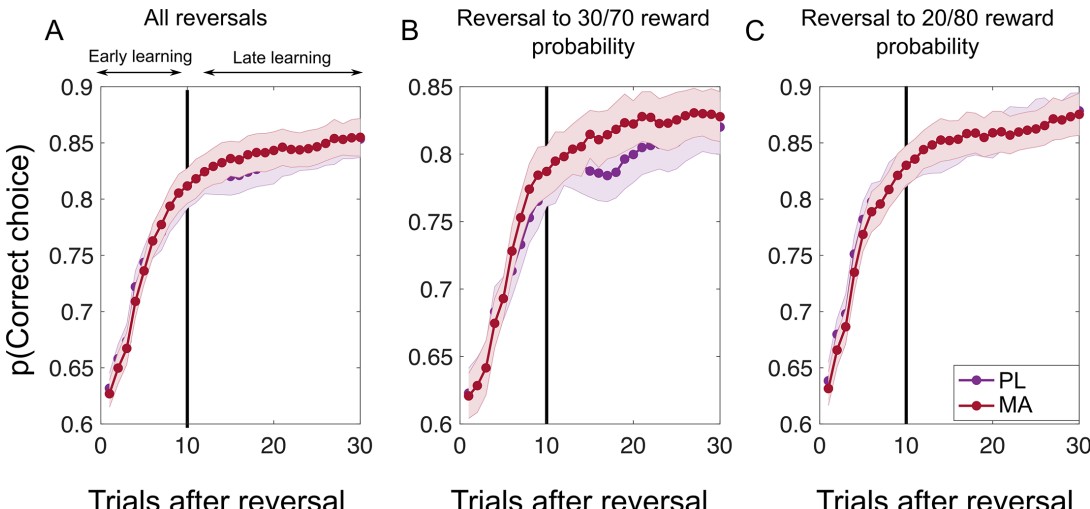

**Appendix 5—figure 2.** Learning curves after reversals full sample. Figure shows learning curves after all reversals (**A**), reversals to high reward probability uncertainty (**B**), and reversals to low reward probability uncertainty (**C**) for the whole sample. Vertical black lines divide learning into early and late stages as suggested by the Bai–Perron multiple break point test. Paired-sample *t*-test revealed no drug-related difference for all reversals during early learning (0.72 (0.01) vs. 0.72 (0.01); *t*(108) = –0.02, p = 0.98, *d* < 0.01) and late learning (0.83 (0.01) vs. 0.84 (0.01); *t*(108) = –0.80, p = 0.42, *d* = 0.04). Similarly, there were no significant differences in both learning stages for
*Appendix 5—figure 2 continued on next page*

*Appendix 5—figure 2 continued*
reversals to low reward probability certainty stimuli (early learning PLA vs. MPH: 0.68 (0.01) vs. 0.69 (0.01); *t*(108) = –0.92, p = 0.35, *d* = 0.08; late learning PLA vs. MPH: 0.80 (0.01) vs. 0.81 (0.01); *t*(108) = –1.48, p = 0.14, *d* = 0.10) or to low reward probability certainty stimuli (early learning PLA vs. MPH: 0.74 (0.01) vs. 0.73 (0.01); *t*(108) = 0.87, p = 0.38, *d* = 0.06; late learning PLA vs. MPH: 0.85 (0.01) vs. 0.85 (0.01); *t*(108) = –0.02, p = 0.97, *d* < 0.01). Mixed-effects ANOVAs that controlled for session-order effects and whether participants performed the orientation session revealed no significant effects (all p > 0.06). *Note.* PL = placebo; MA = methamphetamine.

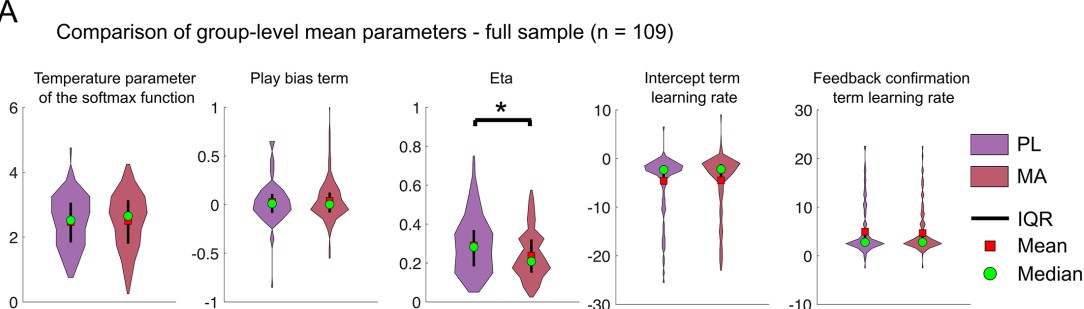

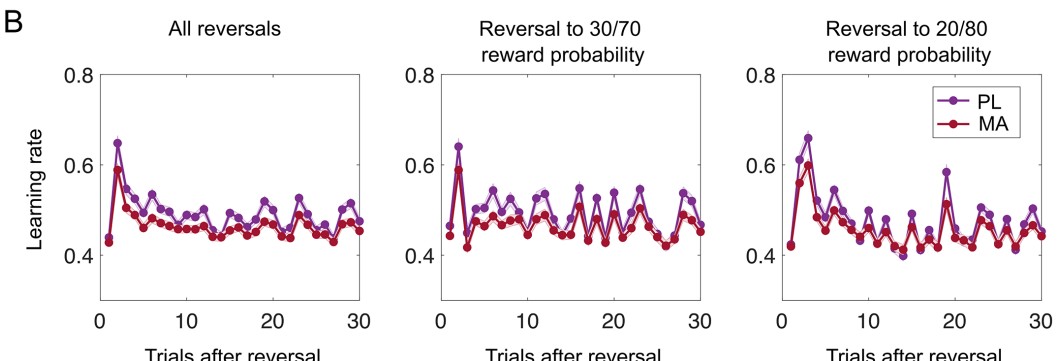

**Appendix 5—figure 3.** Model parameter comparison and learning rate trajectories full sample. (**A**) Here, we compare MPHs effect on best-fitting parameters of the winning model in the full sample (*n* = 109). We found that eta (i.e., the weighting of the effect of the abs. reward prediction error on learning) was reduced under MPHs (eta MPH: 0.23 (0.01) vs. PLA 0.29 (0.01); *t*(108) = –3.05, p = 0.002, *d* = 0.40). Mixed-effects ANOVAs that controlled for session-order effects and whether participants performed the orientation session revealed this effect did not depend on these cofactors. No other condition differences emerged. (**B**) Learning rate trajectories after reversal derived from the computational model. As in the reduced sample, MPH appears to be associated with reduced learning rate dynamics in the full sample too. Specifically, variability in learning rate (average individual SD of learning rate) tended to be reduced in the MPH condition both during early and late stages of learning across all reversals (early PLA: 0.19 (0.01) vs. 0.18 (0.01); *t*(108) = 1.89, p = 0.06, *d* = 0.24; late PLA: 0.17 (0.01) vs. MPH: 0.16 (0.01); *t*(108) = 1.77, p = 0.08, *d* = 0.23) and reversals to high reward probability uncertainty (early PLA: 0.18 (0.01) vs. 0.16 (0.01); *t*(108) = 1.74, p = 0.08, *d* = 0.22; late PLA: 0.18 (0.01) vs. MPH: 0.16 (0.01); *t*(108) = 1.82, p = 0.07, *d* = 0.24). Condition differences became most evident in reversals to low reward probability uncertainty (early PLA: 0.19 (0.01) vs. MPH: 0.16 (0.01); *t*(108) = 2.18, p = 0.03, *d* = 0.28; late PLA: 0.18 (0.01) vs. MPH: 0.16 (0.01); *t*(108) = 1.93, p = 0.05, *d* = 0.24). Control analyses revealed that these effects were independent of session order and orientation session. *Note.* PL = placebo; MA = methamphetamine.

