## [Editor Report · eLife Assessment]

This manuscript reports effects of a single dose of methamphetamine vs placebo on a probabilistic reversal learning task with different levels of noise, in a large group of young healthy volunteers. The paper is well written and the methods are rigorous. The findings are **important** and have theoretical or practical implications beyond a single a subfield. The strength of the evidence is **convincing**, with the methods, data, and analyses broadly supporting the claims in the paper, which are sufficiently qualified given the lack of a significant effect of the binary baseline performance variable, and the nonlinear effect of individual differences in baseline performance.

---

## [Referee Report · Reviewer #1 (Public review)]

The authors examine how probabilistic reversal learning is affected by dopamine by studying the effects of methamphetamine (MA) administration. Based on prior evidence that the effects of pharmacological manipulation depend on baseline neurotransmitter levels, they hypothesized that MA would improve learning in people with low baseline performance. They found this effect, and specifically found that MA administration improved learning in noisy blocks, by reducing learning from misleading performance, in participants with lower baseline performance. The authors then fit participants' behavior to a computational learning model and found that an eta parameter, responsible for scaling learning rate based on previously surprising outcomes, differed in participants with low baseline performance on and off MA.

Questions:

(1) It would be helpful to confirm that the observed effect of MA on the eta parameter is responsible for better performance in low baseline performers. If performance on the task is simulated for parameters estimated for high and low baseline performers on and off MA, does the simulated behavior capture the main behavioral differences shown in Figure 3?

(2) In Figure 4C, it appears that the main parameter difference between low and high baseline performance is inverse temperature, not eta. If MA is effective in people with lower baseline DA, why is the effect of MA on eta and not IT?

Also, this parameter is noted as temperature but appears to be inverse temperature as higher values are related to better performance. The exact model for the choice function is not described in the methods.

Comments on revisions:

Thanks to the authors for their thorough responses and revisions.One typo to note: in the Methods, the "drug effects" paragraph is repeated.

---

## [Referee Report · Reviewer #2 (Public review)]

Summary:

Kirschner and colleagues test whether methamphetamine (MA) alters learning rate dynamics in a validated reversal learning task. They find evidence that MA can enhance performance for low-performers, and that the enhancement reflects a reduction in the degree to which these low-performers dynamically up-regulate their learning rates when they encounter unexpected outcomes. The net effect is that poor performers show more volatile learning rates (e.g. jumping up when they receive misleading feedback), when the environment is actually stable, undermining their performance over trials.

Strengths:

The study has multiple strengths, including a large sample size, placebo control, double-blind randomized design, and rigorous computational modeling of a validated task. Additionally, the analytic methods are rigorous and offer new types of analyses for people interested in exploring learning as a function of dynamically changing volatility.

Weaknesses:

The limitations, which are acknowledged, include that the drug they use, methamphetamine, can influence multiple neuromodulatory systems including catecholamines and acetylcholine, all of which have been implicated in learning rate dynamics. They also do not have any independent measures of any of these systems, so it is impossible to know which is having an effect.

Another limitation which they should acknowledge is that the fact that participants were aware of having different experiences in the drug sessions means that their blinding was effectively single-blind (to the experimenters) and not double-blind. That said, the authors do provide some evidence that subjective effects of drugs (e.g. arousal, mood, etc.) did not drive differences in performance.

Comments on revisions:

The authors have done an outstanding job responding to, and allaying my prior concerns about their analyses.

---

## [Author Response]

The following is the authors’ response to the original reviews.

**Public Reviews:**

**Reviewer #1 (Public review):**
The authors examine how probabilistic reversal learning is affected by dopamine by studying the effects of methamphetamine (MA) administration. Based on prior evidence that the effects of pharmacological manipulation depend on baseline neurotransmitter levels, they hypothesized that MA would improve learning in people with low baseline performance. They found this effect, and specifically found that MA administration improved learning in noisy blocks, by reducing learning from misleading performance, in participants with lower baseline performance. The authors then fit participants' behavior to a computational learning model and found that an eta parameter, responsible for scaling learning rate based on previously surprising outcomes, differed in participants with low baseline performance on and off MA.Questions:(1) It would be helpful to confirm that the observed effect of MA on the eta parameter is responsible for better performance in low baseline performers. If performance on the task is simulated for parameters estimated for high and low baseline performers on and off MA, does the simulated behavior capture the main behavioral differences shown in Figure 3?

We thank the reviewer for this suggestion. We agree that the additional simulation provides valuable confirmation of the effect of methamphetamine (MA) on the eta parameter and subsequent choice behavior. Using individual maximum likelihood parameter estimates, we simulated task performance and confirmed that the simulated behavior reflects the observed mean behavioral differences. Specifically, the simulation demonstrates that MA increases performance later in learning for stimuli with less predictable reward probabilities, particularly in subjects with low baseline performance (mean ± SD: simPL low performance: 0.69 ± 0.01 vs. simMA low performance: 0.72 ± 0.01; t(46) = -2.00, p = 0.03, d = 0.23).

We have incorporated this analysis into the manuscript. Specifically, we added a new figure to illustrate these findings and updated the text accordingly. Below, we detail the changes made to the manuscript.

From the manuscript page 12, line 25:

“Sufficiency of the model was evaluated through posterior predictive checks that matched behavioral choice data (see Figure 4D-F and Figure 5) and model validation analyses (see Supplementary Figure 2). Specifically, using individual maximum likelihood parameter estimates, we simulated task performance and confirmed that MA increases performance later in learning for stimuli with less predictable reward probabilities, particularly in subjects with low baseline performance (Figure 5A; mean ± SD: simPL low performance: 0.69 ± 0.01 vs. simMA low performance: 0.72 ± 0.01; t(46) = -2.00, p = 0.03, d = 0.23).”

(2) In Figure 4C, it appears that the main parameter difference between low and high baseline performance is inverse temperature, not eta. If MA is effective in people with lower baseline DA, why is the effect of MA on eta and not IT?

Thank you for raising this important point. It is correct that the primary difference between the low and high baseline performance groups in the placebo session lies in the inverse temperature (mean(SD); low baseline performance: 2.07 (0.11) vs. high baseline performance: 2.95 (0.07); t(46) = -5.79, p = 5.8442e-07, d = 1.37). However, there is also a significant difference in the eta parameter between these groups during the placebo session (low baseline performance: 0.33 (0.02) vs. baseline performance: 2.07 (0.11243) vs. high baseline performance: 0.25 (0.02); t(46) = 2.59, p = 0.01, d = 0.53).

Interestingly, the difference in eta is resolved by MA (mean(SD); low baseline performance: 0.24 (0.02) vs. high baseline performance: 0.23 (0.02); t(46) = 0.39, p = 0.70, d = 0.08), while the difference in inverse temperature remains unaffected (mean(SD); low baseline performance: 2.16 (0.11) vs. high baseline performance: 2.99 (0.08); t(46) = -5.38, p < .001, d = 1.29). Moreover, we checked the distribution of the inverse temperature estimates on/offdrug to ensure the absent drug effect is not driven by outliers. Here, we do not observe any descriptive drug effect (see Author response image 1). Additionally, non-parametric tests indicate no drug effect (Wilcoxon signed-rank test; across groups: zval = -0.59; p = 0.55; low baseline performance: zval = -0.54; p = 0.58; high baseline performance: zval = -0.21; p = 0.83).

**Author response image 1. sa3fig1:** Inverse temperature distribution on/off drug suggest that this parameter is not affected by the drug. Inverse temperature for low (blue points) and high (yellow points) baseline performer tended to be not affected by the drug effect (Wilcoxon signed-rank test; across groups: zval = -0.59; p = 0.55; low baseline performance: zval = -0.54; p = 0.58; high baseline performance: zval = -0.21; p = 0.83).

This pattern of results might suggests that MA specifically affects eta but not other parameters like the inverse temperature, pointing to a selective influence on a single computational mechanism. To verify this conclusion, we extended the winning model by allowing each parameter in turn to be differentially estimated for MA and placebo, while keeping other parameters fixed to the group (low and high baseline performance) mean estimates of the winning model fit to chocie behaviour of the placebo session.

These control analyses confirmed that MA does not affect inverse temperature in either the low baseline performance group or the high baseline performance group. Similarly, MA did not affect the play bias or learning rate intercept parameter. Yet, it did affect eta in the low performer group (see supplementary table 1 reproduced below).

Taken together, our data suggest that only the parameter controlling dynamic adjustments of the learning rate based on recent prediction errors, eta, was affected by our pharmacological manipulation and that the paremeters of our models did not trade off. A similar effect has been observed in a previous study investigating the effects of catecholaminergic drug administration in a probabilistic reversal learning task (Rostami Kandroodi et al., 2021). In that study, the authors demonstrated that methylphenidate influenced the inverse learning rate parameter as a function of working memory span, assessed through a baseline cognitive task. Similar to our findings, they did not observe drug effects on other parameters in their model including the inverse temperature.

We have updated the section of the manuscript where we discuss the difference in inverse temperature between low and high performers in the task. From the manuscript (page 19, line 13):

“While eta seemed to account for the differences in the effects of MA on performance in our low and high performance groups, it did not fully explain all performance differences across the two groups (see Figure 1C and Figure 7A/B). When comparing other model parameters between low and high baseline performers across drug sessions, we found that high baseline performers displayed higher overall inverse temperatures (2.97(0.05) vs. 2.11 (0.08); t(93) = 7.94, p < .001, d = 1.33). This suggests that high baseline performers displayed higher transfer of stimulus values to actions leading to better performance (as also indicated by the positive contribution of this parameter to overall performance in the GLM). Moreover, they tended to show a reduced play bias (-0.01 (0.01) vs. 0.04 (0.03); t(93) = -1.77, p = 0.08, d = 0.26) and increased intercepts in their learning rate term (-2.38 (0.364) vs. -6.48 (0.70); t(93) = 5.03, p < .001, d = 0.76). Both of these parameters have been associated with overall performance (see Figure 6A). Thus, overall performance difference between high and low baseline performers can be attributed to differences in model parameters other than eta. However, as described in the previous paragraph, differential effects of MA on performance on the two groups were driven by eta.

This pattern of results suggests that MA specifically affects the eta parameter while leaving other parameters, such as the inverse temperature, unaffected. This points to a selective influence on a single computational mechanism. To verify this conclusion, we extended the winning model by allowing each parameter, in turn, to be differentially estimated for MA and PL, while keeping the other parameters fixed at the group (low and high baseline performance) mean estimates of the winning model for the placebo session. These control analyses confirmed that MA affects only the eta parameter in the low-performer group and that there is no parameter-trade off in our model (see Supplementary Table 1). A similar effect was observed in a previous study investigating the effects of catecholaminergic drug administration on a probabilistic reversal learning task (Rostami Kandroodi et al., 2021). In that study, methylphenidate was shown to influence the inverse learning rate parameter (i.e., decay factor for previous payoffs) as a function of working memory span, assessed through a baseline cognitive task. Consistent with our findings, no drug effects were observed on other parameters in their model, including the inverse temperature.”

Additionally, we summarized the results in a supplementary table:

Also, this parameter is noted as temperature but appears to be inverse temperature as higher values are related to better performance. The exact model for the choice function is not described in the methods.

We thank the reviewer for bringing this to our attention. The reviewer is correct that we intended to refer to the inverse temperature. We have corrected this mistake throughout the manuscript and added information about the choice function to the methods section.

From the manuscript (page 37, line 3):

On each trial, this value term was transferred into a “biased” value term (*𝑉𝐵*(*𝑋𝑡*) = *𝐵𝑝𝑙𝑎𝑦* + *𝑄𝑡*(*𝑋𝑡*), where *𝐵𝑝𝑙𝑎𝑦* is the play bias term) and converted into action probabilities (P(play|(*𝑉𝐵 play)*(*𝑡*)(*𝑋𝑡*); P(pass|*𝑉𝐵 pass)*(*𝑡*)(*𝑋𝑡*)) using a softmax function with an inverse temperature (𝛽):\begin{document}$$\displaystyle \left.\mid V_{B \text { play }}(t)\left(X_{t}\right)\right)=\frac{\exp \left(V_{\mathrm{B} \text { play }}(t)\left(X_{t}\right) \cdot \beta\right)}{\exp \left(V_{\text {play }}(t)\left(X_{t}\right) \cdot \beta\right)+\exp \left(V_{\text {pass }}(t)\left(X_{t}\right) \cdot \beta\right)}$$\end{document}\begin{document}$$\displaystyle \mathrm{P}\left(\operatorname{pass} \mid V_{B \text { pass }}(t)\left(X_{t}\right)\right)=\frac{\exp \left(V_{\text {pass }}(t)\left(X_{t}\right) \cdot \beta\right)}{\exp \left(V_{\text {play }}(t)\left(X_{t}\right) \cdot \beta\right)+\exp \left(V_{\text {pass }}(t)\left(X_{t}\right) \cdot \beta\right)}$$\end{document}

**Reviewer #1 (Recommendations for the authors):**
(1) Given that the task was quite long (700+ trials), were there any fatigue effects or changes in behavior over the course of the task?

To address the reviewer comment, we regressed each participant single-trial log-scaled RT and accuracy (binary variable reflecting whether a participant displayed stimulus-appropriate behavior on each trial) onto the trial number as a proxy of time on task. Individual participants’ t-values for the time on task regressor were then tested on group level via two-sided t-tests against zero and compared across sessions and baseline performance groups. The results of these two regression models are shown in the supplementary table 2 and raw data splits in supplementary figure S7. Results demonstrate that the choice behavior was not systematically affected over the course of the task. This effect was not different between low and high baseline performers and not affected by the drug. In contrast, participants’ reaction time decreased over the course of the task and this speeding was enhanced by MA, particularly in the low performance group.

We added the following section to the supplementary materials and refer to this information in the task description section of the manuscript (page 35, line 26):

“Time-on-Task Effects

Given the length of our task, we investigated whether fatigue effects or changes in behavior occurred over time. Specifically, we regressed each participant's single-trial log-scaled reaction times (RT) and accuracy (a binary variable reflecting whether participants displayed stimulus-appropriate behavior on each trial) onto trial number, which served as a proxy for time on task. The resulting t-values for the time-on-task regressor were analyzed at the group level using two-sided t-tests against zero and compared across sessions and baseline performance groups. The results of these regression models are presented in Supplementary Table S2, with raw data splits shown in Supplementary Figure S3.

Our findings indicate that choice behavior was not systematically affected over the course of the task. This effect did not differ between low and high baseline performers and was not influenced by the drug. In contrast, reaction times decreased over the course of the task, with this speeding effect being enhanced by MA, particularly in the low-performance group.”

(2) Figure 5J is hard to understand given the lack of axis labels on some of the plots. Also, the scatter plot is on the left, not the right, as stated in the legend.

We agree that this part of the figure was difficult to understand. To address this issue, we have separated it from Figure 5, added axis labels for clarity, and reworked the figure caption.

(3) The data and code were not available for review.

Thank you for pointing this out. The data and code are now made publicly available onGitHub: https://github.com/HansKirschner/REFIT_Chicago_public.git

We updated the respective section in the manuscript:

Data Availability StatementAll raw data and analysis scripts can be accessed at:https://github.com/HansKirschner/REFIT_Chicago_public.git

**Reviewer #2 (Public review):**
Summary:Kirschner and colleagues test whether methamphetamine (MA) alters learning rate dynamics in a validated reversal learning task. They find evidence that MA can enhance performance for low-performers and that the enhancement reflects a reduction in the degree to which these low-performers dynamically up-regulate their learning rates when they encounter unexpected outcomes. The net effect is that poor performers show more volatile learning rates (e.g. jumping up when they receive misleading feedback), when the environment is actually stable, undermining their performance over trials.Strengths:The study has multiple strengths including large sample size, placebo control, double-blind randomized design, and rigorous computational modeling of a validated task.Weaknesses:The limitations, which are acknowledged, include that the drug they use, methamphetamine, can influence multiple neuromodulatory systems including catecholamines and acetylcholine, all of which have been implicated in learning rate dynamics. They also do not have any independent measures of any of these systems, so it is impossible to know which is having an effect.Another limitation that the authors should acknowledge is that the fact that participants were aware of having different experiences in the drug sessions means that their blinding was effectively single-blind (to the experimenters) and not double-blind. Relatedly, it is difficult to know whether subjective effects of drugs (e.g. arousal, mood, etc.) might have driven differences in attention, causing performance enhancements in the low-performing group. Do the authors have measures of these subjective effects that they could include as covariates of no interest in their analyses?

We thank the reviewer for highlighting this complex issue. ‘Double blind’ may refer to masking the identity of the drug before administration, or to the subjects’ stated identifications after any effects have been experienced. In our study, the participants were told that they might receive a stimulant, sedative or placebo on any session, so before the sessions their expectations were blinded. After receiving the drug, most participants reported feeling stimulant-like effects on the drug session, but not all of them correctly identified the substance as a stimulant. We note that many subjects identified placebo as ‘sedative’. The Author response image 2 indicates how the participants identified the substance they received.

**Author response image 2. sa3fig2:** Substance identification.

We share the reviewer’s interest in the extent to which mood effects of drugs are correlated with the drugs’ other effects, including cognitive function. To address this in the present study, we compared the subjective responses to the drug in participants who were low- or highperformers at baseline on the task. The low- and high baseline performers did not differ in their subjective drug effects, including ‘feel drug’ or stimulant-like effects (see Figure 1 from the mansucript reproduced below; peak change from baseline scores for feel drug ratings ondrug: low baseline performer: 48.36(4.29) vs. high baseline performer: 47.21 (4.44); t(91) = 0.18, p = 0.85, d = 0.03; ARCI-A score: low baseline performer: 4.87 (0.43) vs. high baseline performer: 4.00 (0.418); t(91) = 1.43, p = 0.15, d = 0.30). Moreover, task performance in the drug session was not correlated with the subjective effects (peak “feel drug” effect: r(94) = 0.09, p = 0.41; peak “stimulant like” effect: r(94) = -0.18, p = 0.07).

We have added details of these additional analyses to the manuscript. Since there were no significant differences in subjective drug effects between low- and high-baseline performers, and these effects were not systematically associated with task performance, we did not include these measurements as covariates in our analyses. Furthermore, as both subjective measurements indicate a similar pattern, we have chosen not to report the ARCI-A effects in the manuscript.

From the manuscript (page 6, line 5ff):

“Subjective drug effectsMA administration significantly increased ‘feel drug effect’ ratings compared to PL, at 30, 50, 135, 180, and 210 min post-capsule administration (see Figure 1; Drug x Time interaction F(5,555) = 38.46, p < 0.001). In the MA session, no differences in the ‘feel drug effect’ were observed between low and high baseline performer, including peak change-from-baseline ratings (rating at 50 min post-capsule: low baseline performer: 48.36(4.29) vs. high baseline performer: 47.21 (4.44); t(91) = 0.18, p = 0.85, d = 0.03; rating at 135 min post-capsule: low baseline performer: 37.27 (4.15) vs. high baseline performer: 45.38 (3.84); t(91) = 1.42, p = 0.15, d = 0.29).”

**Reviewer #2 (Recommendations for the authors):**
I was also concerned about the distinctions between the low- and high-performing groups. It is unclear why, except for simplicity of presentation, they chose to binarize the sample into high and low performers. I would like to know if the effects held up if they analyzed interactions with individual differences in performance and not just a binarized high/low group membership. If the individual difference interactions do not hold up, I would like to know the authors' thoughts on why they do not.

Thank you for raising this important issue. We chose a binary discretization of baseline performance to simplify the analysis and presentation. However, we acknowledge that this simplification may limit the interpretability of the results.

To address the reviewer’s concern, we conducted additional linear mixed-effects model (LMM) analyses, focusing on the key findings reported in the manuscript. See supplementary materials section “Linear mixed effects model analyses for key findings”

From the manuscript (page 30, line 4ff):

“Methamphetamine performance enhancement depends on initial task performance

Another key finding of the current study is that the benefits of MA on performance depend on the baseline task performance. Specifically, we found that MA selectively improved performance in participants that performed poorly in the baseline session. However, it should be noted, that all the drug x baseline performance interactions, including for the key computational eta parameter did not reach the statistical threshold, and only tended towards significance. We used a binary discretization of baseline performance to simplify the analysis and presentation. To parse out the relationship between methamphetamine effects and baseline performance into finer level of detail, we conducted additional linear mixed-effects model (LMM) analyses using a sliding window regression approach (see supplementary results and supplementary figure S4 and S5). A key thing to notice in the sliding regression results is that, while each regression reveals that drug effects depend on baseline performance, they do so non-linearly, with most variables of interest showing a saturating effect at low baseline performance levels and the strongest slope (dependence on baseline) at or near the median level of baseline performance, explaining why our median splits were able to successfully pick up on these baseline-dependent effects. Together, these results suggest that methamphetamine primarily affects moderately low baseline performer. It is noteworthy to highlight again that we had a separate baseline measurement from the placebo session, allowing us to investigate baseline-dependent changes while avoiding typical concerns in such analyses like regression to the mean (Barnett et al., 2004). This design enhances the robustness of our baseline-dependent effects.”

See supplementary materials section “Linear mixed effects model analyses for key findings”

Perhaps relatedly, in multiple analyses, the authors point out that there are drug effects for the low-performance group, but not the high-performance group. This could reflect the well-documented baseline-dependency effect of catecholamergic drugs. However, it might also reflect the fact that the high-performance group is closer to their ceiling. So, a performance-enhancement drug might not have any room to make them better. Note that their results are not consistent with inverted-U-like effects, previously described, where high performers actually get worse on catecholaminergic drugs.Given that the authors have the capacity to simulate performance as a function of parameter values, they could specifically simulate how much better performance could get if their high-performance group all moved proportionally closer to optimal levels of the parameter eta. On the basis of that analysis do they have any evidence that they had the power to detect an effect in the high performance group? If not, they should just acknowledge that ceiling effects might have played a role for high performers.

We agree with the reviewer's interpretation of the results. First, when plotting overall task performance and the probability of correct choices in the high outcome noise condition—the condition where we observe the strongest drug-induced performance enhancement—we find minimal performance variation among high baseline performers. In both testing sessions, high baseline performers cluster around optimal performance, with little evidence of drug-induced changes (see Supplementary Figure 6).

Furthermore, performance simulations using (a) optimal eta values and (b) observed eta values from the high baseline performance group reveal only a small, non-significant performance difference (points optimal eta: 701.91 (21.66) vs. points high performer: 694.47 (21.71); t(46) = 2.84, p = 0.07, d = 0.059).

These results suggest that high baseline performers are already near optimal performance, limiting the potential for drug-related performance improvements. We have incorporated this information into the manuscript (page 30, line 24ff).

“It is important to note, that MA did not bring performance of low baseline performers to the level of performance of high baseline performers. We speculate that high performers gained a good representation of the task structure during the orientation practice session, taking specific features of the task into account (change point probabilities, noise in the reward probabilities). This is reflected in a large signal to noise ratio between real reversals and misleading feedback. Because the high performers already perform the task at a near-optimal level, MA may not further enhance performance (see Supplementary Figure S6 for additional evidence for this claim). Intriguingly, the data do not support an inverted-u-shaped effect of catecholaminergic action (Durstewitz & Seamans, 2008; Goschke & Bolte, 2018) given that performance of high performers did not decrease with MA. One could speculate that catecholamines are not the only factor determining eta and performance. Perhaps high performers have a generally more robust/resilient decision-making system which cannot be perturbed easily. Probably one would need even higher doses of MA (with higher side effects) to impair their performance.”

Finally, I am confused about why participants are choosing correctly at higher than 50% on the first trial after a reversal (see Figure 3)? How could that be right? If it is not, does this mean that there is a pervasive error in the analysis pipeline?

Thank you for pointing this out. The observed pattern is an artifact of the smoothing (±2 trials) applied to the learning curves in Figure 3. Below, we reproduce the figure without smoothing.

Additionally, we confirm that the probability of choosing the correct response is not above chance level (t-test against chance):• All reversals: t(93)=1.64,p=0.10,d=0.17, 99% CI[0.49,0.55]• Reversal to low outcome noise: t(93)=1.67,p=0.10,d=0.17, 99% CI [0.49,0.56]• Reversal to high outcome noise: t(93)=0.87,p=0.38,d=0.09, 99% CI [0.47,0.56]

We have amended the caption of Figure 3 accordingly. Moreover, we included an additional figure in this revision letter (Author response image 4) showing a clear performance drop to approximately 50% correct choices across all sessions, indicating random-choice behavior at the point of reversal. Notably, this performance is slightly better than expected (i.e., the inverse of pre-reversal performance). One possible explanation is that participants developed an expectation of the reversal, leading to increased reversal behaviour around reversals.

**Author response image 3. sa3fig3:** Learning curves after reversals suggest that methamphetamine improves learning performance in phases of less predictable reward contingencies in low baseline performer. Top panel of the Figure shows learning curves after all reversals (A), reversals to stimuli with less predictable reward contingencies (B), and reversals to stimuli with high reward probability certainty (C). Bottom panel displays the learning curves stratified by baseline performance for all reversals (D), reversals to stimuli with less predictable reward probabilities (E), and reversals to stimuli with high reward probability certainty (F). Vertical black lines divide learning into early and late stages as suggested by the Bai-Perron multiple break point test. Results suggest no clear differences in the initial learning between MA and PL. However, learning curves diverged later in the learning, particular for stimuli with less predictable rewards (B) and in subjects with low baseline performance (E). Note. PL = Placebo; MA = methamphetamine; Mean/SEM = line/shading.

**Author response image 4. sa3fig4:** Adaptive behavior following reversals. Each graph shows participants' performance (i.e., stimulus-appropriate behavior: playing good stimuli with 70/80% reward probability and passing on bad stimuli with 20/30% reward probability) around reversals for the (A) orientation session, (B) placebo session, and (C) methamphetamine session. Trial 0 corresponds to the trial when reversals occurred, unbeknownst to participants. Participants' performance exhibited a fast initial adaptation to reversals, followed by a slower, late-stage adjustment to the new stimulus-reward contingencies, eventually reaching a performance plateau. Notably, we observe a clear performance drop to approximately 50% correct choices across all sessions, indicating random-choice behavior at the point of reversal. This performance is slightly better than expected (i.e., the inverse of pre-reversal performance). One possible explanation is that participants developed an expectation of the reversal, leading to increased reversal behaviour around reversals.

Minor comments:(1) I'm unclear on what the analysis in 6E tells us. What does it mean that the marginal effect of eta on performance predicts changes in performance? Also, if multiple parameters besides eta (e.g. learning rate) are strongly related to actual performance, why should it be that only marginal adjustments to eta in the model anticipate actual performance improvements when marginal adjustments to other model parameters do not?

We agree that these simulations are somewhat difficult to interpret and have therefore decided to omit these analyses from the manuscript. Our key point was that individuals who benefited the most from methamphetamine were those who exhibited the most advantageous eta adjustments in response to it. We believe this is effectively illustrated by the example individual shown in Figure 8D.

(2) Does the vertical black line in Figure 1 show when the tasks were completed, as it says in the caption, or when the task starts, as it indicates in the figure itself?

Apologies for the confusion. There was a mistake in the figure caption—the vertical line indicates the time when the task started (60 minutes post-capsule intake). We have corrected this in the figure caption.

(3) The marginally significant drug x baseline performance group interaction does not support strong inferences about differences in drug effects on eta between groups...

We agree and have added information on this limitation to the Discussion. Additionally, we have addressed the complex relationship between drug effects and baseline performance in the supplementary analyses, as detailed in our previous response regarding the binary discretization of baseline performance.

(4) Should lines 10-11 on page 12 say "We did not find drug-related differences in any other model parameters..."?

Thank you for bringing this grammatical error to our attention. We have corrected it.

(5) It would be good to confirm that the effect of MA on p(Correct after single MFB) does not have an opposite sign from the effect of MA on p(Correct after double MFB). I'm guessing the effect after single is just weak, but it would be good to confirm they are in the same direction so that we can be confident the result is not picking up on spurious relationships after two misleading instances of feedback.

We confirm that the direction of the effect between eta and p(Correct after single MFB) is similar to p(Correct after double MFB). First, we see a similar negative association between p(Correct after single MFB) and eta (r(94) = -.26, p = 0.01). Similarly there was a descriptive increase in p(Correct after single MFB) for low baseline performer on- vs. off-drug (p(Correct after single MFB): low baseline performance PL: 0.71 (0.02) vs. low baseline performance MA:0.73 (0.02); t(46) = 1.27, p = 0.20, d = 0.17).

(6) "implemented equipped" seems like a typo on page 16, line 26

Thank you for bringing this typo to our attention. We have corrected it.

**Reviewing Editor (Public Review):**
Summary:In this well-written paper, a pharmacological experiment is described in which a large group of volunteers is tested on a novel probabilistic reversal learning task with different levels of noise, once after intake of methamphetamine and once after intake of placebo. The design includes a separate baseline session, during which performance is measured. The key result is that drug effects on learning rate variability depend on performance in this separate baseline session.The approach and research question are important, the results will have an impact, and the study is executed according to current standards in the field. Strengths include the interventional pharmacological design, the large sample size, the computational modeling, and the use of a reversal-learning task with different levels of noise.(i) One novel and valuable feature of the task is the variation of noise (having 70-30 and 8020 conditions). This nice feature is currently not fully exploited in the modeling of the task and the data. For example, recently reported new modeling approaches for disentangling two types of uncertainty (stochasticity vs volatility) could be usefully leveraged here (by Piray and Daw, 2021, Nat Comm). The current 'signal to noise ratio' analysis that is targeting this issue relies on separately assessing learning rates on true reversals and learning rates after misleading feedback, in a way that is experimenter-driven. As a result, this analysis cannot capture a latent characteristic of the subject's computational capacity.

We thank the reviewing editor for the positive evaluation of our work and the suggestion to leverage new modeling approaches. In the light of the Piray/Daw paper, it is noteworthy, that the choice behavior of the low performance group in our sample mimics the behavior of their lesioned model, in which stochasticity is assumed to be small and constant. Specifically, low performers displayed higher learning rates, particularly in high outcome noise phases in our task. One possible interpretation of this choice pattern is that they have problems to distinguish volatility and noise. Consistently, surprising outcomes may get misattributed to volatility instead of stochasticity resulting in increased learning rates and overadjustments to misleading outcomes. This issue particularly surfaces in phases of high stochasticity in our task. Interestingly, methamphetamine seems to reduce this misattribution. In an exploratory analysis, we fit two models to our task structure using modified code provided by the Piray and Daw paper. The control model made inference about both the volatility and stochasticity. A key assumption of the model is, that the optimal learning rate increases with volatility and decreases with stochasticity. This is because greater volatility raises the likelihood that the underlying reward probability has changed since the last observation, increasing the necessity of relying on new information. In contrast, higher stochasticity reduces the relative informativeness of the new observation compared to prior beliefs about the underlying reward probability. The lesioned model assumed stochasticity to be small and constant. We show the results of this analyses in Figure 9 and Supplementary Figure S5 and S6. Interestingly, we found that the inability to make inference about stochasticity leads to misestimation of volatility, particularly for high outcome noise phases (Figure 9A-B). Consistently, this led to reduced sensitivity of the learning rate to volatility (i.e., the first ten trials after reversals). The model shows similar behaviour to our low performer group, with reduced accuracy in later learnings stages for stimuli with high outcome noise (Figure 9D). Finally, when we fit simulated data from the two models to our model, we see increased eta parameter estimates for the lesioned model. Together, these results may hint towards an overinterpretation of stochasticity in low performers of our task and that methamphetamine has beneficial effects for those individuals as it reduced the oversensitivity to volatility. It should be noted however, that we did not fit these models to our choice behaviour directly as this implementation is beyond the scope of our current study. Yet, our exploratory analyses make testable predictions for future research into the effect of catecholamines on the inference of volatility and stochasticity.

We incorporated information on these explorative analyses to the manuscript and supplementary material.

Form the result section (page 23, line 12ff):

“Methamphetamine may reduce misinterpretation of high outcome noise in low performers

In our task, outcomes are influenced by two distinct sources of noise: process noise (volatility) and outcome noise (stochasticity). Optimal learning rate should increase with volatility and decrease with stochasticity. Volatility was fairly constant in our task (change points around every 30-35 trials). However, misleading feedback (i.e., outcome noise) could be misinterpreted as indicating another change point because participants don’t know the volatility beforehand. Strongly overinterpreting outcome noise as change points will hinder building a correct estimate of volatility and understanding the true structure of the task. Simultaneously estimating volatility and stochasticity poses a challenge, as both contribute to greater outcome variance, making outcomes more surprising. A critical distinction, however, lies in their impact on generated outcomes: volatility increases the autocorrelation between consecutive outcomes, whereas stochasticity reduces it. Recent computational approaches have successfully utilised this fundamental difference to formulate a model of learning based on the joint estimation of stochasticity and volatility (Piray & Daw, 2021; Piray & Daw, 2024). They report evidence that humans successfully dissociate between volatility and stochasticity with contrasting and adaptive effects on learning rates, albeit to varying degrees. Interestingly they show that hypersensitivity to outcome noise, often observed in anxiety disorders, might arise from a misattribution of the outcome noise to volatility instead of stochasticity resulting in increased learning rates and overadjustments to misleading outcomes. It is noteworthy, that we observed a similar hypersensitivity to high outcome noise in low performers in our task that is partly reduced by MA. In an exploratory analysis, we fit two models to our task structure using modified code provided by Piray and Daw (2021) (see Methods for formal Description of the model). The control model inferred both the volatility and stochasticity. The lesioned model assumed stochasticity to be small and constant. We show the results of this analyses in Figure 9 and Supplementary Figure S7 and S8. We found that the inability to make inference about stochasticity, leads to misestimation of volatility, particularly for high outcome noise phases (Figure 9A-B). Consistently, this led to reduced sensitivity of the learning rate to volatility (i.e., the first ten trials after reversals). The model shows similar behaviour to our low performer group, with reduced accuracy in later learning stages for stimuli with high outcome noise (Figure 9D). Finally, when we fit simulated data from the two models to our model, we see increased eta parameter estimates for the lesioned model. Together, these results may hint towards an overinterpretation of stochasticity in low performer of our task and that MA has beneficial effects for those individuals as it reduced the oversensitivity to volatility. It should be noted however, that we did not fit these models to our choice behaviour directly as this implementation is beyond the scope of our current study. Yet, our exploratory analyses make testable predictions for future research into the effect of catecholamines on the inference of volatility and stochasticity.”

From the discussion (page 28, line 15ff):

“Exploratory simulation studies using a model that jointly estimates stochasticity and volatility (Piray & Daw, 2021; Piray & Daw, 2024), revealed that MA might reduce the oversensitivity to volatility.”

See methods section “Description of the joint estimation of stochasticity and volatility model “

(ii) An important caveat is that all the drug x baseline performance interactions, including for the key computational eta parameter did not reach the statistical threshold, and only tended towards significance.

We agree and have added additional analyses on the issue. See also our response to reviewer 2. There is a consistent effect for low-medium baseline performance. We toned done the reference to low baseline performance but still see strong evidence for a baseline dependency of the drug effect.

From the manuscript (page 30, line 4ff):

“Methamphetamine performance enhancement depends on initial task performance

Another key finding of the current study is that the benefits of MA on performance depend on the baseline task performance. Specifically, we found that MA selectively improved performance in participants that performed poorly in the baseline session. However, it should be noted, that all the drug x baseline performance interactions, including for the key computational eta parameter did not reach the statistical threshold, and only tended towards significance. We used a binary discretization of baseline performance to simplify the analysis and presentation. To parse out the relationship between methamphetamine effects and baseline performance into finer level of detail, we conducted additional linear mixed-effects model (LMM) analyses using a sliding window regression approach (see supplementary results and supplementary figure S4 and S5). A key thing to notice in the sliding regression results is that, while each regression reveals that drug effects depend on baseline performance, they do so non-linearly, with most variables of interest showing a saturating effect at low baseline performance levels and the strongest slope (dependence on baseline) at or near the median level of baseline performance, explaining why our median splits were able to successfully pick up on these baseline-dependent effects. Together, these results suggest that methamphetamine primarily affects moderately low baseline performer. It is noteworthy to highlight again that we had a separate baseline measurement from the placebo session, allowing us to investigate baseline-dependent changes while avoiding typical concerns in such analyses like regression to the mean (Barnett et al., 2004). This design enhances the robustness of our baseline-dependent effects.”

(iii) Both the overlap and the differences between the current study and previous relevant work (that is, how this goes beyond prior studies in particular Rostami Kandroodi et al, which also assessed effects of catecholaminergic drug administration as a function of baseline task performance using a probabilistic reversal learning task) are not made explicit, particularly in the introduction.

Thank you for raising this point. We have added information of the overlap and differences between our paper and the Rostami Kondoodi et al paper to the introduction and disscussion.

In the intoduction we added a sentence to higlight the Kondoordi findings (page 3, line 24ff).

For example, Rostami Kandroodi et al. (2021) reported that the re-uptake blocker methylphenidate did not alter reversal learning overall, but preferentially improved performance in participants with higher working memory capacity.”

In our Discussion, we go back to this paper, and say how our findings are and are not consistent with their findings (page 32, line 16ff).

Our findings can be contrasted to those of Rostami Kandroodi et al. (2021), who examined effects of methylphenidate on a reversal learning task, in relation to baseline differences on a cognitive task. Whereas Rostami Kandroodi et al. (2021) found that the methylphenidate improved performance mainly in participants with higher baseline working memory performance, we found that methamphetamine improved the ability to dynamically adjust learning from prediction errors to a greater extent in participants who performed poorly-tomedium at baseline. There are several possible reasons for these apparently different findings. First, MA and methylphenidate differ in their primary mechanisms of action: MPH acts mainly as a reuptake blocker whereas MA increases synaptic levels of catecholamines by inhibiting the vesicular monoamine transporter 2 (VMAT2) and inhibiting the enzyme monoamine oxidase (MAO). These differences in action could account for differential effects on cognitive tasks. Second, the tasks used by Rostami Kandroodi et al. (2021) and the present study differ in several ways. The Rostami Kandroodi et al. (2021) task assessed responses to a single reversal event during the session whereas the present study used repeated reversals with probabilistic outcomes. Third, the measures of baseline function differed in the two studies: Rostami Kandroodi et al. (2021) used a working memory task that was not used in the drug sessions, whereas we used the probabilistic learning task as both the baseline measure and the measure of drug effects. Further research is needed to determine which of these factors influenced the outcomes.”performance effects, but this is not true in the general sense, given that an accumulating number of studies have shown that the effects of drugs like MA depend on baseline performance on working memory tasks, which often but certainly not always correlates positively with performance on the task under study.

We recognize that there is a large body of research reporting that the effects of stimulant drugs are related to baseline performance, and we have adjusted our wording in the Discussion accordingly. At the same time, numerous published studies report acute effects of drugs without considering individual differences in responses, including baseline differences in task performance.

**Reviewing Editor (Recommendations for the Authors):**
(i) To leverage recently reported new modeling approaches for disentangling two types of uncertainty (stochasticity vs volatility) might be usefully leveraged (Piray and Daw, 2021, Nat Comm) to help overcome the shortcomings of the 'signal-to-noise ratio' analysis performed here (learning rates on true reversals minus learning rates after misleading feedback) which is experimenter-driven, and thus cannot capture a latent characteristic of the subject's computational capacity.

Please see our previous response.

(ii) To highlight more explicitly the fact that various of the key drug x baseline performance interactions did not reach the statistical threshold.

Please see our previous responses to this issue.

(iii) To make more explicit, in the introduction, both the overlap and the differences between the current study and previous relevant work (that is, how this goes beyond prior study in particular Rostami Kandroodi et al, which also assessed effects of catecholaminergic drug administration as a function of baseline task performance using a probabilistic reversal learning task).

Please see our previous response.

(iv) To revise and tone down, in the discussion section, the statement about novelty, that the existing literature has, to date, overlooked baseline performance effects.

Please see our previous response.

(v) It is unclear why the data from the 4th session (under some other sedative drug, which is not mentioned) are not reported. I recommend justifying the details of this manipulation and the decision to omit the report of those results. By analogy 4 other tasks were administered in the current study, but not described. Is there a protocol paper, describing the full procedure?

Thank you for pointing this out. We added additional information to the method section. We are analysing the other cognitive measures in relation to the brain imaging data obtained on sessions 3 and 4. Therefore we argue, that these are beyond the scope of the present paper. We did not administer any sedative drug. However, participants were informed during orientation that they might receive a stimulant, sedative, or placebo on any testing session to maintain blinding of their expectations before each session.

“Design. The results presented here were obtained from the first two sessions of a larger foursession study (https://clinicaltrials.gov/ ID number NCT04642820). During the latter two sessions of the larger study, not reported here, participants participated in two fMRI scans. During the two 4-h laboratory sessions presented here, healthy adults received methamphetamine (20 mg oral; MA) or placebo (PL), in mixed order under double-blind conditions. One hour after ingesting the capsule they completed the 30-min reinforcement reversal learning task. The primary comparisons were on acquisition and reversal learning parameters of reinforcement learning after MA vs PL. Secondary measures included subjective and cardiovascular responses to the drug.”

“Orientation session. Participants attended an initial orientation session to provide informed consent, and to complete personality questionnaires. They were told that the purpose of the study was to investigate the effects of psychoactive drugs on mood, brain, and behavior. To reduce expectancies, they were told that they might receive a placebo, stimulant, or sedative/tranquilizer. However, participants only received methamphetamine and placebo. They agreed not to use any drugs except for their normal amounts of caffeine for 24 hours before and 6 hours following each session. Women who were not on oral contraceptives were tested only during the follicular phase (1-12 days from menstruation) because responses to stimulant drugs are dampened during the luteal phase of the cycle (White et al., 2002). Most participants (N=97 out of 113) completed the reinforcement learning task during the orientation session as a baseline measurement. This measure was added after the study began. Participants who did not complete the baseline measurement were omitted from the analyses presented in the main text. We run the key analyses on the full sample (n=109). This sample included participants who completed the task only on the drug sessions. When controlling for session order and number (two vs. three sessions) effects, we see no drug effect on overall performance and learning. Yet, we found that eta was also reduced under MA in the full sample, which also resulted in reduced variability in the learning rate (see supplementary results for more details).”

“Drug sessions. The two drug sessions were conducted in a comfortable laboratory environment, from 9 am to 1 pm, at least 72 hours apart. Upon arrival, participants provided breath and urine samples to test for recent alcohol or drug use and pregnancy (CLIAwaived Inc,Carlsbad, CAAlcosensor III, Intoximeters; AimStickPBD, hCG professional, Craig Medical Distribution). Positive tests lead to rescheduling or dismissal from the study. After drug testing, subjects completed baseline mood measures, and heart rate and blood pressure were measured. At 9:30 am they ingested capsules (PL or MA 20 mg, in color-coded capsules) under double-blind conditions. Oral MA (Desoxyn, 5 mg per tablet) was placed in opaque size 00 capsules with dextrose filler. PL capsules contained only dextrose. Subjects completed the reinforcement learning task 60 minutes after capsule ingestion. Drug effects questionnaires were obtained at multiple intervals during the session. They completed other cognitive tasks not reported here. Participants were tested individually and were permitted to relax, read or watch neutral movies when they were not completing study measures.”

(vi) Some features of the model including the play bias parameter require justification, at least by referring to prior work exploring these features.

We have added information to justify the features of the model.

Form the method section:

“The base model (M1) was a standard Q-learning model with three parameters: (1) an inverse temperature parameter of the softmax function used to convert trial expected values to action probabilities, (2) a play bias term that indicates a tendency to attribute higher value to gambling behavior (Jang et al., 2019), ….

The two additional learning rate terms—feedback confirmation and modality—were added to the model set, as these factors have been shown to influence learning in similar tasks (Kirschner et al., 2023; Schüller et al., 2020).”

Literature

Doucet, A., & Johansen, A. M. (2011). A tutorial on particle filtering and smoothing: fifteen years later. Oxford University Press.

Durstewitz, D., & Seamans, J. K. (2008). The dual-state theory of prefrontal cortex dopamine function with relevance to catechol-o-methyltransferase genotypes and schizophrenia. Biol Psychiatry, 64(9), 739-749. https://doi.org/10.1016/j.biopsych.2008.05.015

Gamerman, D., dos Santos, T. R., & Franco, G. C. (2013). A NON-GAUSSIAN FAMILY OF STATE-SPACE MODELS WITH EXACT MARGINAL LIKELIHOOD. Journal of Time Series Analysis, 34(6), 625-645.

https://doi.org/10.1111/jtsa.12039

Goschke, T., & Bolte, A. (2018). A dynamic perspective on intention, conflict, and volition: Adaptive regulation and emotional modulation of cognitive control dilemmas. In Why people do the things they do: Building on Julius Kuhl’s contributions to the psychology of motivation and volition. (pp. 111-129). Hogrefe. https://doi.org/10.1027/00540-000

Jang, A. I., Nassar, M. R., Dillon, D. G., & Frank, M. J. (2019). Positive reward prediction errors during decision-making strengthen memory encoding. Nature Human Behaviour, 3(7), 719-732. https://doi.org/10.1038/s41562-019-0597-3

Jenkins, D. G., & Quintana-Ascencio, P. F. (2020). A solution to minimum sample size for regressions. PLoS One, 15(2), e0229345. https://doi.org/10.1371/journal.pone.0229345

Kirschner, H., Nassar, M. R., Fischer, A. G., Frodl, T., Meyer-Lotz, G., Froböse, S., Seidenbecher, S., Klein, T. A., & Ullsperger, M. (2023). Transdiagnostic inflexible learning dynamics explain deficits in depression and schizophrenia. Brain, 147(1), 201-214. https://doi.org/10.1093/brain/awad362

Maris, E., & Oostenveld, R. (2007). Nonparametric statistical testing of EEG- and MEG-data. Journal of Neuroscience Methods, 164(1), 177-190.

https://doi.org/10.1016/j.jneumeth.2007.03.024

Morean, M. E., de Wit, H., King, A. C., Sofuoglu, M., Rueger, S. Y., & O'Malley, S. S. (2013). The drug effects questionnaire: psychometric support across three drug types. Psychopharmacology (Berl), 227(1), 177-192. https://doi.org/10.1007/s00213-0122954-z

Murphy, K., & Russell, S. (2001). Rao-Blackwellised particle filtering for dynamic Bayesian networks. In Sequential Monte Carlo methods in practice (pp. 499-515). Springer. Piray, P., & Daw, N. D. (2020). A simple model for learning in volatile environments. PLoS Comput Biol, 16(7), e1007963. https://doi.org/10.1371/journal.pcbi.1007963

Piray, P., & Daw, N. D. (2021). A model for learning based on the joint estimation of stochasticity and volatility. Nature Communications, 12(1), 6587. https://doi.org/10.1038/s41467-021-26731-9

Piray, P., & Daw, N. D. (2024). Computational processes of simultaneous learning of stochasticity and volatility in humans. Nat Commun, 15(1), 9073. https://doi.org/10.1038/s41467-024-53459-z

Rostami Kandroodi, M., Cook, J. L., Swart, J. C., Froböse, M. I., Geurts, D. E. M., Vahabie, A. H., Nili Ahmadabadi, M., Cools, R., & den Ouden, H. E. M. (2021). Effects of methylphenidate on reinforcement learning depend on working memory capacity. Psychopharmacology (Berl), 238(12), 3569-3584. https://doi.org/10.1007/s00213021-05974-w

Schüller, T., Fischer, A. G., Gruendler, T. O. J., Baldermann, J. C., Huys, D., Ullsperger, M., & Kuhn, J. (2020). Decreased transfer of value to action in Tourette syndrome. Cortex, 126, 39-48. https://doi.org/10.1016/j.cortex.2019.12.027

West, M. (1987). On scale mixtures of normal distributions. Biometrika, 74(3), 646-648. https://doi.org/10.1093/biomet/74.3.646

White, T. L., Justice, A. J., & de Wit, H. (2002). Differential subjective effects of Damphetamine by gender, hormone levels and menstrual cycle phase. Pharmacol Biochem Behav, 73(4), 729-741.